 **eLIFE**

# Expansion of the fatty acyl reductase gene family shaped pheromone communication in Hymenoptera

**Michal Tupec[1,2†], Aleš Buček[1,3†]\*, Václav Janoušek[4], Heiko Vogel[5], Darina Prchalová[1], Jiří Kindl[1], Tereza Pavlíčková[1], Petra Wenzelová[1], Ullrich Jahn[1], Irena Valterová[1], Iva Pichová[1]\***

[1]Institute of Organic Chemistry and Biochemistry of the Czech Academy of Sciences, Prague, Czech Republic; [2]Department of Biochemistry, Faculty of Science, Charles University, Prague, Czech Republic; [3]Okinawa Institute of Science and Technology Graduate University, Okinawa, Japan; [4]Department of Zoology, Faculty of Science, Charles University, Prague, Czech Republic; [5]Department of Entomology, Max Planck Institute for Chemical Ecology, Jena, Germany

**\*For correspondence:**
ales.bucek@oist.jp (AšBč);
iva.pichova@uochb.cas.cz (IPá)

[†]These authors contributed equally to this work

**Competing interests:** The authors declare that no competing interests exist.

**Abstract** Fatty acyl reductases (FARs) are involved in the biosynthesis of fatty alcohols that serve a range of biological roles. Insects typically harbor numerous FAR gene family members. While some FARs are involved in pheromone biosynthesis, the biological significance of the large number of FARs in insect genomes remains unclear.

Using bumble bee (Bombini) FAR expression analysis and functional characterization, hymenopteran FAR gene tree reconstruction, and inspection of transposable elements (TEs) in the genomic environment of FARs, we uncovered a massive expansion of the FAR gene family in Hymenoptera, presumably facilitated by TEs. The expansion occurred in the common ancestor of bumble bees and stingless bees (Meliponini). We found that bumble bee FARs from the expanded FAR-A ortholog group contribute to the species-specific pheromone composition. Our results indicate that expansion and functional diversification of the FAR gene family played a key role in the evolution of pheromone communication in Hymenoptera.

DOI: https://doi.org/10.7554/eLife.39231.001

## Introduction

Accumulation of DNA sequencing data is greatly outpacing our ability to experimentally assess the function of the sequenced genes, and most of these genes are expected to never be functionally characterized (*Koonin, 2005*). Important insights into the evolutionary processes shaping the genomes of individual species or lineages can be gathered from predictions of gene families, gene ortholog groups and gene function. However, direct experimental evidence of the function of gene family members is often unavailable or limited (*Lespinet et al., 2002*; *Demuth et al., 2006*; *Rispe et al., 2008*; *Cortesi et al., 2015*; *Niimura and Nei, 2006*). Gene duplication is hypothesized to be among the major genetic mechanisms of evolution (*Ohno, 1970*; *Zhang, 2003*). Although the most probable evolutionary fate of duplicated genes is the loss of one copy, the temporary redundancy accelerates gene sequence divergence and can result in gene subfunctionalization or neofunctionalization—acquisition of slightly different or completely novel functions in one copy of the gene (*Innan and Kondrashov, 2010*; *Lynch and Conery, 2000*).

The alcohol-forming fatty acyl-CoA reductases (FARs, EC 1.2.1.84) belong to a multigene family that underwent a series of gene duplications and subsequent gene losses, pseudogenizations and possibly functional diversification of some of the maintained copies, following the birth-and-death

**eLife digest** Many insects release chemical signals, known as sex pheromones, to attract mates over long distances. The pheromones of male bumble bees, for example, contain chemicals called fatty alcohols. Each species of bumble bee releases a different blend of these chemicals, and even species that are closely related may produce very different 'cocktails' of pheromones.

The enzymes that make fatty alcohols are called fatty acyl reductases (or FARs for short). Any change to a gene that encodes one of these enzymes could change the final mix of pheromones produced. This in turn could have far-reaching effects for the insect, and in particular its mating success. Over time, these changes could even result in new species. Yet no one has previously looked into how the genes for FAR enzymes have evolved in bumble bees, or how these genes might have shaped the evolution of this important group of insects.

Tupec, Buček et al. set out to learn what genetic changes led the males of three common species of bumble bees to make dramatically different mixes of pheromones. Comparing the genetic information of bumble bees with that of other insects showed that the bumble bees and their close relatives, stingless bees, often had extra copies of genes for certain FAR enzymes. Inserting some of these genes into yeast cells caused the yeast to make the correct blend of bumble bee pheromones, confirming that these genes did indeed produce the mixture of chemicals in these signals.

Further, detailed analysis of the bumble bees' genetic information revealed many genetic sequences, called transposable elements, close to the genes for the FAR enzymes. Transposable elements make the genetic material less stable; they can be 'cut' or 'copied and pasted' in multiple locations and often cause other genes to be duplicated or lost. Tupec et al. concluded that these transposable elements led to a dramatic increase in the number of genes for FAR enzymes in a common ancestor of bumble bees and stingless bees, ultimately allowing a new pheromone 'language' to evolve in these insects.

These results add to our understanding of the chemical and genetic events that influence what chemicals insects use to communicate with each other. Tupec, Buček et al. also hope that a better knowledge of the enzymes that insects use to make pheromones could have wide applications. Other insects – including pest moths – use a similar mixture of fatty alcohols as pheromones. Artificially produced enzymes, such as FAR enzymes, could thus be used to mass-produce pheromones that may control insect pests.

DOI: https://doi.org/10.7554/eLife.39231.002

model of gene family evolution (*Eirín-López et al., 2012*). FARs exhibit notable trends in gene numbers across organism lineages; there are very few FAR genes in fungi, vertebrates and non-insect invertebrates such as *Caenorhabditis elegans*, whereas plant and insect genomes typically harbor an extensive number of FAR gene family members (*Eirín-López et al., 2012*). FARs are critical for production of primary fatty alcohols, which are naturally abundant fatty acid (FA) derivatives with a wide variety of biological roles. Fatty alcohols are precursors of waxes and other lipids that serve as surface-protective or hydrophobic coatings in plants, insects and other animals (*Wang et al., 2017*; *Cheng and Russell, 2004*; *Jaspers et al., 2014*); precursors of energy-storing waxes (*Metz et al., 2000*; *Teerawanichpan and Qiu, 2010a*; *Teerawanichpan and Qiu, 2012*); and components of ether lipids abundant in the cell membranes of cardiac, nervous and immunological tissues (*Nagan and Zoeller, 2001*).

Additionally, in some insect lineages, FARs were recruited for yet another task—biosynthesis of fatty alcohols that serve as pheromones or pheromone precursors. Moths (Lepidoptera) are the most well-studied model of insect pheromone biosynthesis and have been the subject of substantial research effort related to FARs. Variation in FAR enzymatic specificities is a source of sex pheromone signal diversity among moths in the genus *Ostrinia* (*Lassance et al., 2013*) and is also responsible for the distinct pheromone composition in two reproductively isolated races of the European corn borer *Ostrinia nubilalis* (*Lassance et al., 2010*). Divergence in pheromone biosynthesis can potentially install or strengthen reproductive barriers, ultimately leading to speciation (*Smadja and Butlin, 2009*). However, the biological significance of a large number of insect FAR paralogs remains unclear, as all FARs implicated in moth and butterfly sex pheromone biosynthesis are restricted to a

single clade, indicating that one FAR group was exclusively recruited for pheromone biosynthesis (*Lassance et al., 2010*; *Liénard et al., 2010*; *Liénard et al., 2014*; *Antony et al., 2009*). While more than 20 FARs have been experimentally characterized from 23 moth and butterfly (Lepidoptera) species (*Tupec et al., 2017*), FARs from other insect orders have received far less attention. Single FAR genes have been isolated and experimentally characterized from *Drosophila* (Diptera) (*Jaspers et al., 2014*), the European honey bee (Hymenoptera) (*Teerawanichpan et al., 2010b*) and the scale insect *Ericeus pela* (Hemiptera) (*Hu et al., 2018*). Our limited knowledge of FAR function prevents us from drawing inferences about the biological significance of the FAR gene family expansion in insects.

Bumble bees (Hymenoptera: Apidae) are a convenient experimental model to study insect FAR evolution because the majority of bumble bee species produces fatty alcohols as species-specific components of male marking pheromones (MMPs) (*Ayasse and Jarau, 2014*), which are presumed to be biosynthesized by some of the numerous bumble bee FAR gene family members (*Buček et al., 2016*). Bumble bee males employ MMPs to attract conspecific virgin queens (*Goulson, 2010*). In addition to fatty alcohols, MMPs generally contain other FA derivatives and terpenoid compounds. MMP fatty alcohols consist predominantly of saturated, mono-unsaturated and poly-unsaturated fatty alcohols with 16–18 carbon atoms (*Ayasse and Jarau, 2014*). Pheromone mixtures from three common European bumble bee species, *B. terrestris*, *B. lucorum* and *B. lapidarius*, are representative of the known MMP chemical diversity. Fatty alcohols are the major compounds in MMPs of *B. lapidarius* (hexadecanol and *Z*9-hexadecenol) and accompany electroantennogram-active compounds in *B. terrestris* (hexadecanol, octadecatrienol, octadecenol) and *B. lucorum* (hexadecanol, *Z*9,*Z*12-octadecadienol, *Z*9,*Z*12,*Z*15-octadecatrienol, octadecanol) (*Bergström et al., 1973*; *Kullenberg et al., 1973*; *Kullenberg et al., 1970*; *Urbanová et al., 2001*; *Sobotník et al., 2008*; *Luxová et al., 2003*; *Zácek et al., 2009*).

In our previous investigation of the molecular basis of pheromone diversity in bumble bees, we found that the substrate specificities of fatty acyl desaturases (FADs), enzymes presumably acting upstream of FARs in pheromone biosynthesis (*Tillman et al., 1999*), are conserved across species despite differences in the compositions of their unsaturated FA-derived pheromone components (*Buček et al., 2013*). These findings suggest that the substrate specificity of FADs contributes only partially to the species-specific composition of FA-derived MMPs (*Buček et al., 2013*). The fatty alcohol content in bumble bee MMPs is therefore presumably co-determined by the enzymatic specificity of other pheromone biosynthetic steps, such as FA biosynthesis/transport or FA reduction. Analysis of the *B. terrestris* male labial gland (LG) transcriptome uncovered a remarkably high number of putative FAR paralogs, including apparently expressed pseudogenes, strongly indicating dynamic evolution of the FAR gene family and the contribution of FARs to the LG-localized MMP biosynthesis (*Buček et al., 2016*).

Here, we aimed to determine how the members of the large FAR gene family in the bumble bee lineage contribute to MMP biosynthesis. We sequenced *B. lapidarius* male LG and fat body (FB) transcriptomes and functionally characterized the FAR enzymes, along with FAR candidates from *B. terrestris* and *B. lucorum*, in a yeast expression system. We combined experimental information about FAR enzymatic specificities with quantitative information about bumble bee FAR expression patterns, as well as comprehensive gas chromatography (GC) analysis of MMPs and their FA precursors in the bumble bee male LG, with inference of the hymenopteran FAR gene tree. In addition, we investigated the content of transposable elements (TEs) in the genomic environment of FAR genes in genomes of two bumble bee species, *B. terrestris* and *B. impatiens*. We conclude that a dramatic TE-mediated expansion of the FAR gene family started in the common ancestor of the bumble bee (Bombini: *Bombus*) and stingless bee (Meliponini) lineages, which presumably shaped the pheromone communication in these lineages.

## Results

### Identification of FARs in bumble bee transcriptomes

We sequenced, assembled and annotated male LG and FB transcriptomes of the bumble bee species *B. lapidarius*. The LG is the MMP-producing organ and is markedly enlarged in males, while the FB was used as a reference tissue not directly involved in MMP biosynthesis (*Žáček et al., 2015*).

Searches for FAR-coding transcripts in the LG and FB transcriptomes of *B. lapidarius* and the previously sequenced FB and LG transcriptomes of *B. lucorum* and *B. terrestris* (*Buček et al., 2013*; *Prchalová et al., 2016*) yielded 12, 26 and 16 expressed FAR homologs in *B. lapidarius*, *B. terrestris* and *B. lucorum*, respectively (*Figure 1—figure supplement 1*).

## FAR gene family evolution in hymenoptera

To gain insight into the evolution of FAR gene family in Hymenoptera, we reconstructed a FAR gene tree using predicted FARs from species representing ants, Vespid wasps, parasitoid wasps and several bee lineages (*Figure 1*). We assigned the names FAR-A to FAR-K to 11 FAR ortholog groups that were retrieved as branches with high bootstrap support in the FAR gene tree. These ortholog groups typically encompass one or more FARs from each of the hymenopteran species used in the tree inference, with the exception of apparent species-specific FAR duplications or losses (*Figure 1*). Notably, we identified a massive expansion of the FAR-A ortholog group in the bumble bee and stingless bee (subfamily Meliponini) sister lineages (*Figure 1*, *Figure 2*). The number of FAR homologs is inflated by a large number of predicted FAR genes and FAR transcripts with incomplete protein coding sequences lacking catalytically critical regions, such as the putative active site, NAD(P)$^+$ binding site or substrate binding site (*Figure 1*, *Figure 1—source data 1*). The FAR gene tree also indicates expansion of the FAR-A ortholog group in the ant *Camponotus floridanus* and the mining bee *Andrena vaga*. However, this expansion is not present in two other ant species (*Acromyrmex echinatior* and *Harpegnathos saltator*) and two other mining bee species (Andrenidae: *Camptopoeum sacrum* and *Panurgus dentipes*) (*Figure 1*, *Figure 2*). Several additional expansions of FAR families can be inferred from the FAR gene tree, including extensive FAR-B gene expansion in ants (Formicoidea), along with many more lineage-specific FAR gene duplications and minor expansions.

We also reconstructed a FAR gene tree encompassing FARs from three representatives of non-hymenopteran insect orders—the beetle *Tribolium castaneum*, the moth *Bombyx mori* and the fly *Drosophila melanogaster* (*Figure 1—figure supplement 2*). The only functionally characterized FAR from *D. melanogaster*—Waterproof (NP_651652.2), which is involved in the biosynthesis of a protective wax layer (*Jaspers et al., 2014*)—was placed in the FAR-J ortholog group (*Figure 1—figure supplement 2*). The FAR-G ortholog group includes a FAR gene from *Apis mellifera* with unclear biological function (*Teerawanichpan et al., 2010b*) and a sex pheromone-biosynthetic FAR from *B. mori* (*Moto et al., 2003*) (*Figure 1—figure supplement 2*). In the gene tree, the majority of FAR ortholog groups contain predicted FARs from both hymenopteran and non-hymenopteran insect species, although the bootstrap support is <80% for some FAR groups. The presence of these FAR ortholog groups in representatives of Hymenoptera, Coleoptera, Lepidoptera and Diptera indicates that these groups are ancestral to holometabolous insects. FAR-D is the only FAR group that does not include any non-hymenopteran FARs from our dataset (*Figure 1—figure supplement 2*) and thus presumably represents Hymenoptera-specific FAR gene family expansions. FAR-Ds, however, do not contain the complete set of three catalytically critical regions (i.e. the putative active site, NAD(P)$^+$ binding site and substrate binding site) and their enzymatic role is therefore unclear.

## Genomic organization and TE content

To uncover the details of genetic organization of FAR-A genes, we attempted to analyze the shared synteny of FAR genes in the genomes of *B. terrestris* and *A. mellifera* (*Stolle et al., 2011*). We aligned the *A. melifera* and *B. terrestris* genomes, but we were not able to identify any positional *A. mellifera* homologs of *B. terrestris* FAR-A genes (data not shown). While the majority of FAR genes belonging to the non-FAR-A gene ortholog group localize to the *B. terrestris* genome assembled to linkage groups, most of the *B. terrestris* FAR-A genes localize to unlinked short scaffolds (*Supplementary file 1*). Some of the FAR-A genes in the *B. terrestris* genome are arranged in clusters (*Figure 1—figure supplement 3*). Unfortunately, the genome assembly of *B. impatiens* is not mapped to chromosomes to allow similar analysis.

A genome assembly consisting of short scaffolds is often indicative of a repetitive structure in the assembled genomic region. Our analysis of the distribution of TEs in the vicinity of FAR genes in the *B. terrestris* and *B. impatiens* genomes confirmed that TEs are significantly enriched around FAR-A genes compared to the genome-wide average around randomly selected genes (*Figure 3AC*; *B. terrestris: p < 0.0001*, *B. impatiens: p < 0.0001*). On average, more than 50% of the 10 kb regions

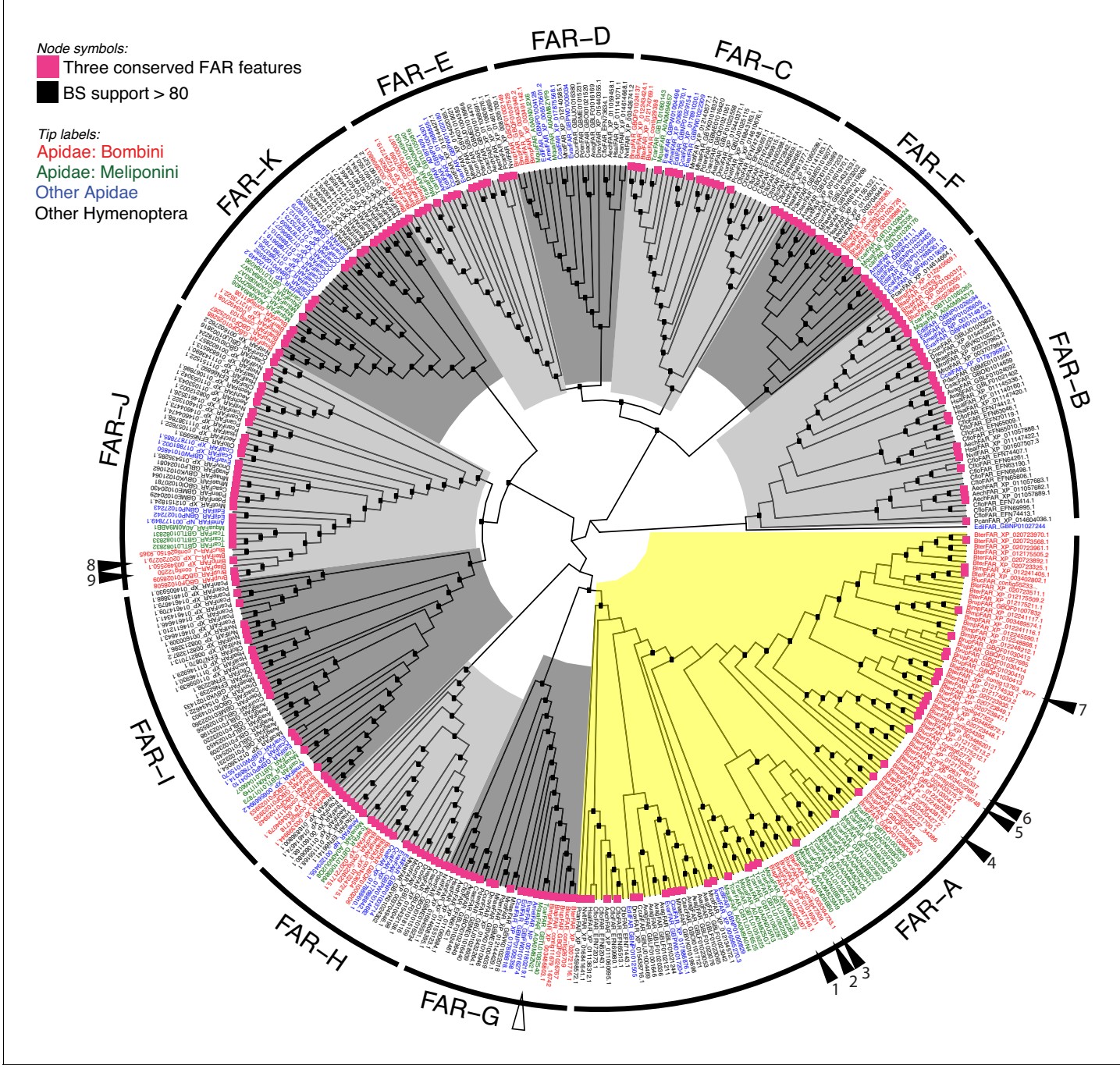

**Figure 1.** Unrooted hymenopteran FAR gene tree. Tree tips are colored according to taxonomy: red, bumble bee FARs (*B. terrestris*, *B. lucorum*, *B. lapidarius*, *B. impatiens*, *B. rupestris*); green, stingless bee FARs (*Tetragonula carbonaria*, *Melipona quadrifasciata*); blue, FARs from other Apidae species (i.e. *A. mellifera*, *Euglossa dilemma*, *Ceratina calcarata*, and *Epeolus variegatus*); and black, FARs from other hymenopteran species. The FAR-A ortholog group is highlighted yellow; other ortholog groups in shades of grey. Functionally characterized bumble bee FARs from this study are indicated by filled triangles and numbered. 1: *Blap*FAR-A1, 2: *Bluc*FAR-A1, 3: *Bter*FAR-A1, 4: *Blap*FAR-A4, 5: *Bluc*FAR-A2, 6: *Bter*FAR-A2, 7: *Blap*FAR-A5, 8: *Bter*FAR-J, and 9: *Blap*FAR-J. The functionally characterized *A. mellifera* FAR is indicated by an empty triangle. Internal nodes highlighted with black boxes indicate bootstrap support >80%. Violet squares at the tree tips indicate FARs for which CDD search yielded all three FAR conserved features—active site, NAD(P)$^+$ binding site and putative substrate binding site (see *Figure 1—source data 1* for complete CDD search results).

DOI: https://doi.org/10.7554/eLife.39231.003

The following source data and figure supplements are available for figure 1:

**Source data 1.** Predicted protein sequence lengths and conserved domains detected in predicted FAR coding regions via Conserved Domain Database search.

*Figure 1 continued on next page*

*Figure 1 continued*
DOI: https://doi.org/10.7554/eLife.39231.009
**Figure supplement 1.** Expression of FARs in male labial gland and male fat body of *Bombus terrestris* (**A**) *B. lucorum* (**B**) and *B. lapidarius* (**C**).
DOI: https://doi.org/10.7554/eLife.39231.004
**Figure supplement 2.** FAR gene tree including non-hymenopteran (*Drosophila melanogaster, Bombyx mori, Tribolium castaneum*) and hymenopteran FARs.
DOI: https://doi.org/10.7554/eLife.39231.005
**Figure supplement 3.** Clusters of FAR-A genes on *B. terrestris* genomic scaffolds Un679 (**A**) and Un989 (**B**).
DOI: https://doi.org/10.7554/eLife.39231.006
**Figure supplement 4.** Protein sequence identities of bumble bee male marking pheromone (MMP)-biosynthetic FAR candidates.
DOI: https://doi.org/10.7554/eLife.39231.007
**Figure supplement 5.** Protein sequence alignment of bumble bee FAR candidates.
DOI: https://doi.org/10.7554/eLife.39231.008

surrounding FAR-A genes are formed by TEs, compared to an average of 10% around randomly selected *B. terrestris* genes. In contrast, the densities of TEs in the vicinity of FAR genes not belonging to the FAR-A group do not differ from the genome-wide average (*Figure 3AC*; *B. terrestris*: $p = 0.793$; *B. impatiens*: $p = 0.880$). Also, there is a statistically significant difference in TE densities between FAR-A genes and non-FAR-A genes (Man-Whitney $U$-test: *B. terrestris*: $U = 167$, $p < 0.0001$; *B. impatiens*: $U = 78$, $p = 0.0002$). Although all major known TE families are statistically enriched in the neighborhood of the FAR-A genes (*Figure 3BD*), the Class I comprising retroid elements contributes considerably to the elevated repeat content around FAR-A genes (*Figure 3*, *Figure 3—source data 1*).

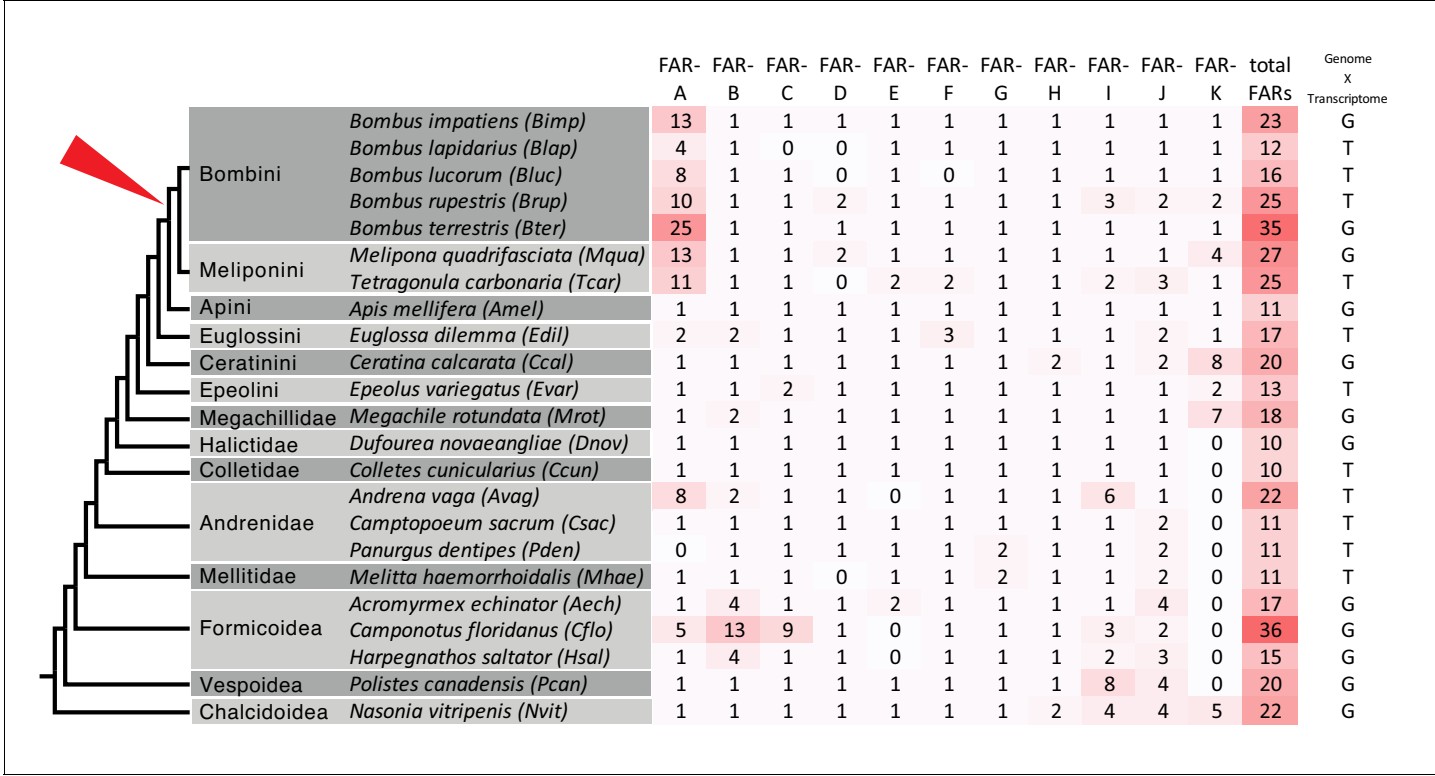

**Figure 2.** Number of predicted FAR genes (transcripts) across hymenopteran lineages and across FAR groups. The schematic phylogenetic tree of Hymenoptera was adapted from *Peters et al. (2017)*. The rightmost column indicates whether FARs were predicted from genome or transcriptome assemblies. The red triangle points to the presumed onset of FAR-A expansion in the common ancestor of bumble bees (Bombini) and stingless bees (Meliponini).
DOI: https://doi.org/10.7554/eLife.39231.010

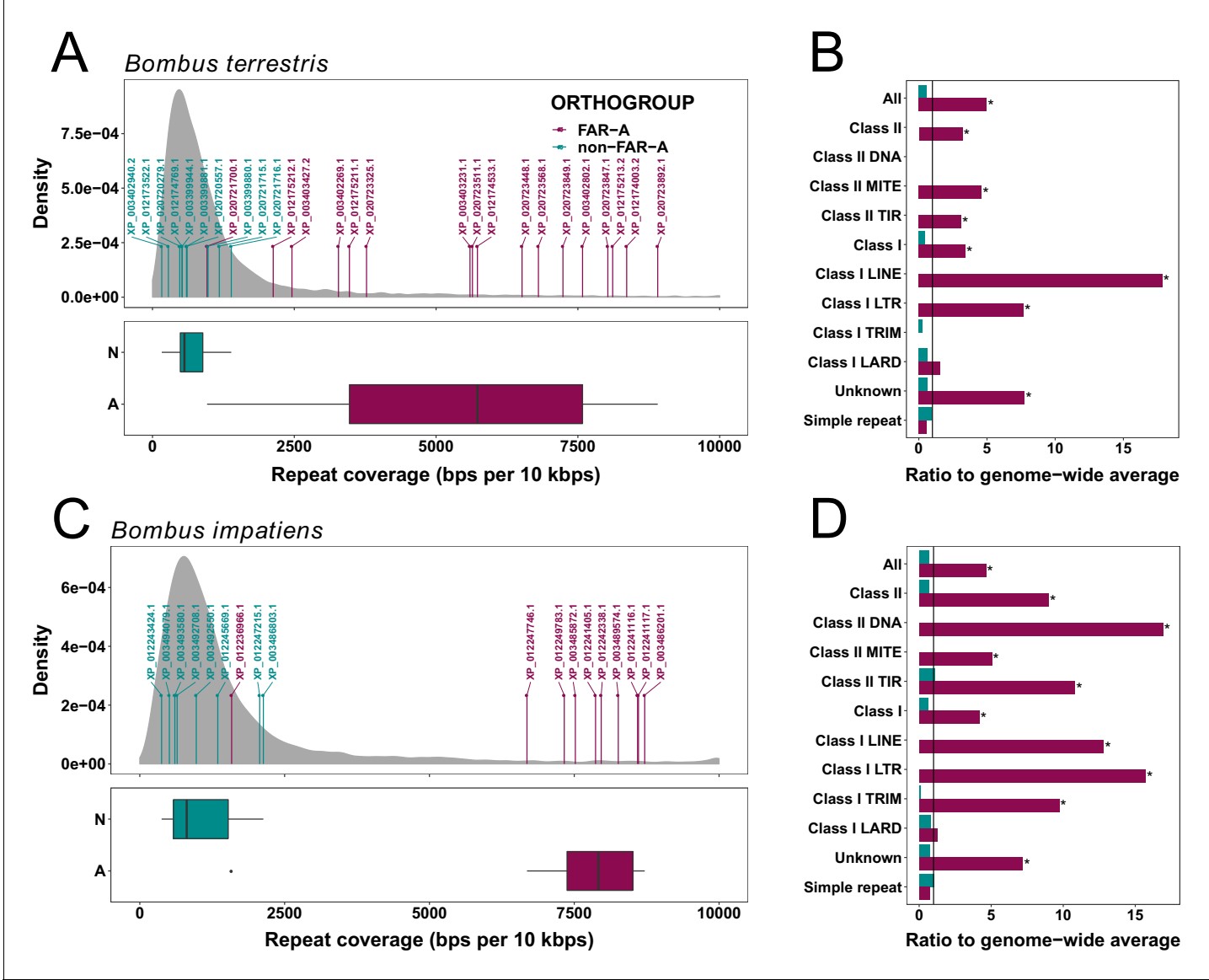

**Figure 3.** Average TE densities in 10 kb windows around groups of *B. terrestris* (A, B) and *B. impatiens* (C, D) genes. (A and C) Distributions of overall TE densities around FAR genes (N, non-FAR-A genes, in green; A, FAR-A genes, in red) with respect to the genome-wide distribution of TE densities around RefSeq genes (in grey) are shown for the *B. terrestris* (A) and *B. impatiens* (C) genomes. (B and D) Densities of individual TE classes and families for FAR-A (in red) and non-FAR-A (in green) genes are depicted for *B. terrestris* (B) and *B. impatiens* (D). The average TE densities for the whole gene group are compared to the genome-wide average. Asterisks indicate statistical significance obtained by permutation test.

DOI: https://doi.org/10.7554/eLife.39231.011

The following source data is available for figure 3:

**Source data 1.** List of TE densities for FAR-A and non-FAR-A genes.

DOI: https://doi.org/10.7554/eLife.39231.012

## Tissue specificity of FAR expression

We selected 10 promising MMP-biosynthetic FAR candidates that were (1) among the 100 most abundant transcripts in the LG and were substantially more abundant in LG than in FB based on RNA-Seq-derived normalized expression values (*Figure 1—figure supplement 1* and ref. (*Buček et al., 2016*)) and (2) included all the predicted catalytically critical regions of FARs—the putative active site, NAD(P)$^+$ binding site and substrate binding site in the protein coding sequence (*Figure 1—source data 1*).

By employing reverse transcription-quantitative PCR (RT-qPCR) on an expanded set of bumble bee tissues, we confirmed that the FAR candidates follow a general trend of overexpression in male LG compared to FB, flight muscle and gut (all from male bumble bees) and virgin queen LG (*Figure 4*, *Figure 4—figure supplement 1*, p < 0.05). Notably, *B. lapidarius* FAR-A1 (*Blap*FAR-A1) and *B. terrestris* FAR-J (*Bter*FAR-J) transcripts are also abundant in virgin queen LG, where they are expressed at levels comparable to those in male LG (*Figure 4—figure supplement 1*).

## Analysis of fatty alcohols and fatty acyls in bumble bee male LG and FB

We performed a detailed analysis of transesterifiable fatty acyls (free FAs and fatty acyls bound in esters) and fatty alcohols in LGs and FBs of 3-day-old *B. lapidarius*, *B. terrestris* and *B. lucorum* males to identify the products and predict the potential FAR substrates in the male LG. In the LGs, we detected 4, 14 and 19 individual fatty alcohol compounds in *B. lapidarius*, *B. lucorum* and *B. terrestris*, respectively (*Figure 5*). A limited number of fatty alcohols (mainly 16:OH, *Z*9,*Z*12-18:OH and *Z*9, *Z*12,*Z*15-18:OH) also were detected in FBs of *B. lucorum* and *B. terrestris*, but at substantially lower abundance than in LGs (*Figure 5—source data 1*).

To assess the apparent *in vivo* specificity of all FARs expressed in LGs and FBs, we calculated the fatty alcohol ratios (see *Equation 1* in Materials and methods), that is the ratios of the quantity of

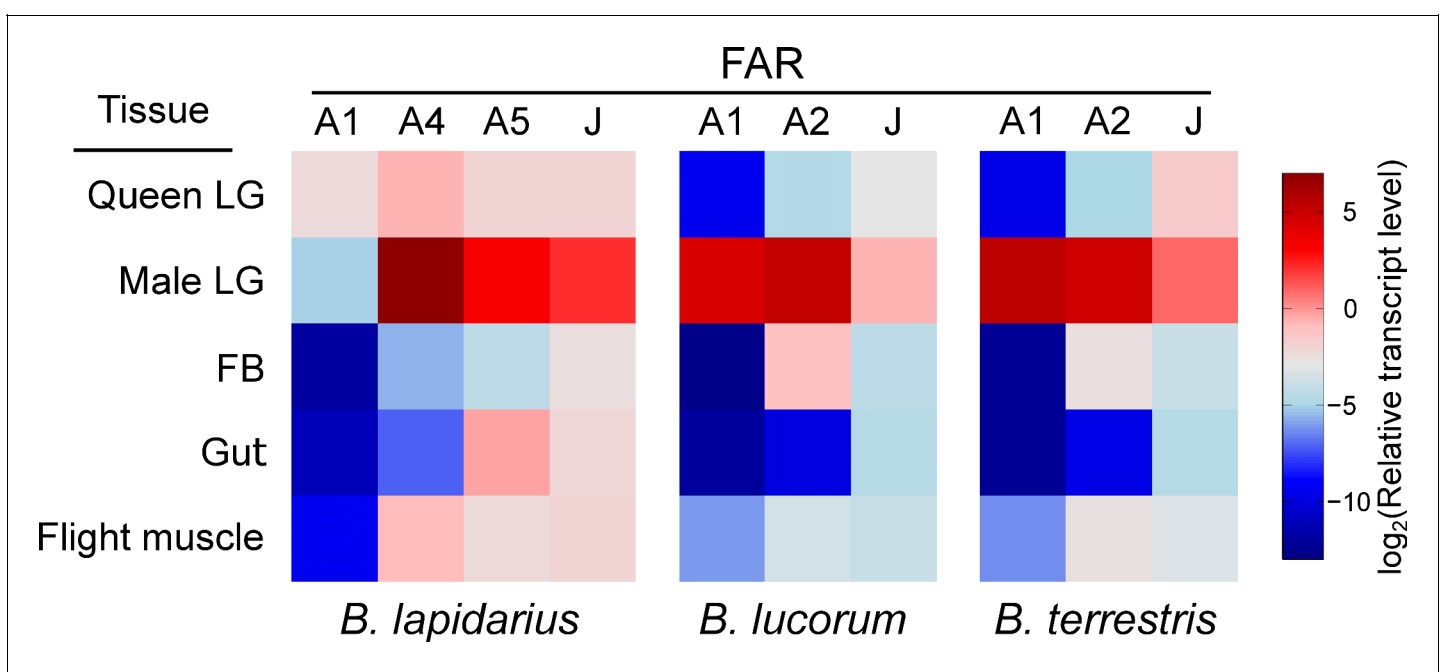

**Figure 4.** Expression pattern of FAR candidates across bumble bee tissues. The FAR transcripts were assayed using quantitative PCR on cDNA from tissues of 3-day-old *B. lapidarius*, *B. lucorum* and *B. terrestris* males and queens (N = 3; queen LG: one biological replicate represents tissues from two queens).

DOI: https://doi.org/10.7554/eLife.39231.013

The following source data and figure supplements are available for figure 4:

**Source data 1.** List of $C_p$ differences for FAR transcripts.

DOI: https://doi.org/10.7554/eLife.39231.016

**Figure supplement 1.** Relative expression of FAR candidates in the tissues of *B. lapidarius*, *B. lucorum* and *B. terrestris* males and queens.

DOI: https://doi.org/10.7554/eLife.39231.014

**Figure supplement 1—source data 1.** List of relative transcript levels and $C_p$ values for FAR candidates.

DOI: https://doi.org/10.7554/eLife.39231.018

**Figure supplement 2.** Relative expression of FAR-A1 and FAR-A1-short in male and queen LGs of *B. lapidarius*.

DOI: https://doi.org/10.7554/eLife.39231.015

**Figure supplement 2—source data 2.** List of relative transcript level, $C_p$ difference and $C_p$ values for FAR-A1 and FAR-A1-short.

DOI: https://doi.org/10.7554/eLife.39231.017

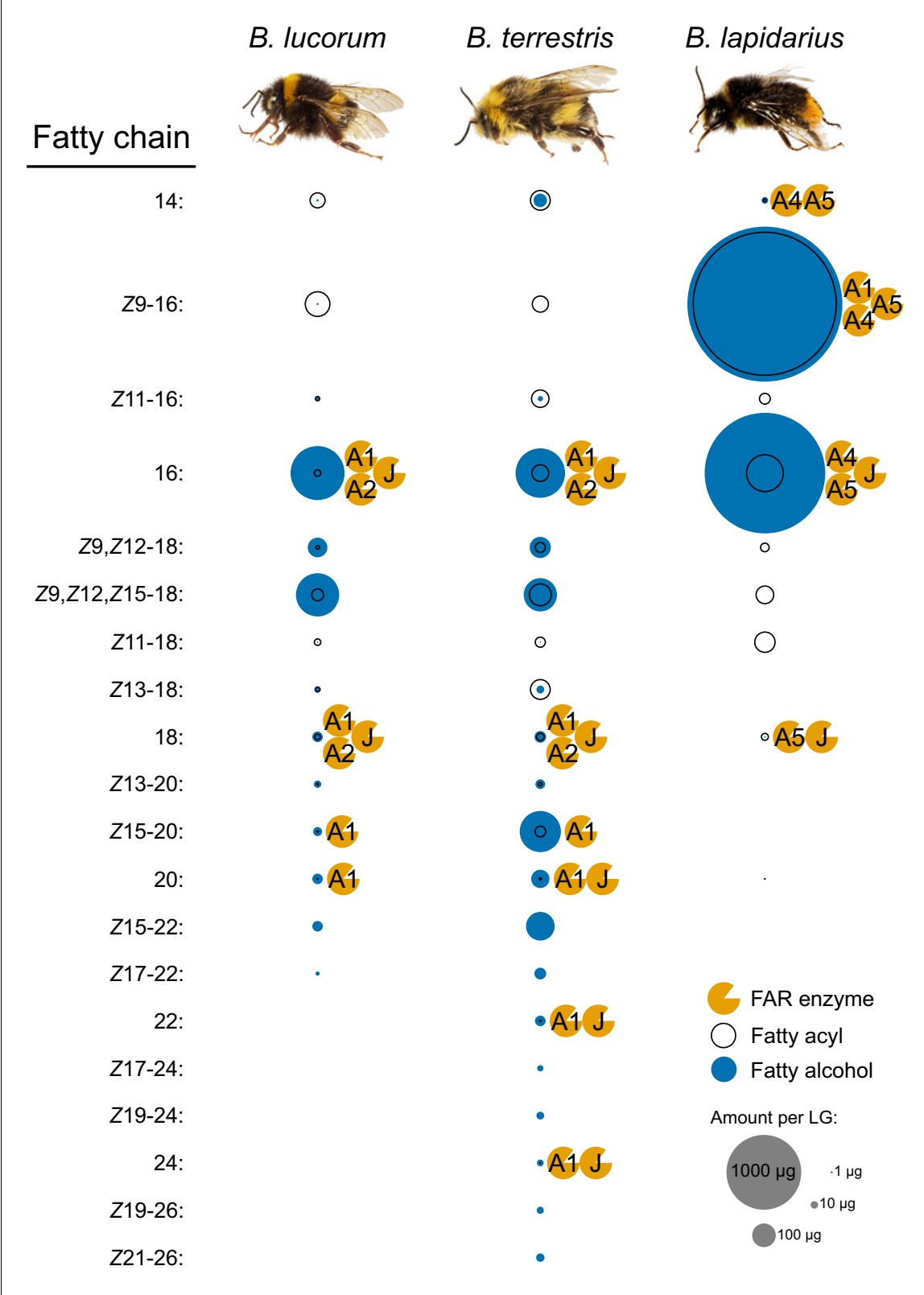

**Figure 5.** Fatty alcohols and fatty acyls of bumble bee male LGs together with the proposed participation of FARs in biosynthesis. The fatty alcohols and fatty acyls (determined as methyl esters) were extracted from LGs of 3-day-old males of *B. lucorum*, *B. terrestris* and *B. lapidarius* (*N* = 3) and quantified by GC. The area (size) of fatty alcohol and fatty acyl circle represents the mean quantity per a single male LG (*Figure 5—source data 1*). The

*Figure 5 continued on next page*

Figure 5 continued

FARs which could be involved in the fatty alcohol biosynthesis based on their specificity are appended to the left side of the corresponding fatty alcohol circle.

DOI: https://doi.org/10.7554/eLife.39231.019

The following source data and figure supplement are available for figure 5:

**Source data 1.** List of fatty alcohol and fatty acyl quantities in LGs and FBs of bumble bee males.

DOI: https://doi.org/10.7554/eLife.39231.021

**Figure supplement 1.** Apparent specificity of FARs in bumble bee LGs and FBs.

DOI: https://doi.org/10.7554/eLife.39231.020

particular fatty alcohol to the quantity of its hypothetical fatty acyl precursors (*Figure 5—figure supplement 1*). These ratios are greater than 50% for most of the fatty alcohols in LGs and even approach 100% for some of the monounsaturated >C20 fatty alcohols, suggesting high overall conversion rates of acyl substrates to alcohols.

## Cloning and functional characterization

The full-length coding regions of the FAR candidates were isolated from male LG cDNA libraries using gene-specific PCR primers (*Supplementary file 2*). In general, the FAR candidates share high to very high protein sequence similarity within each ortholog group (*Figure 1—figure supplement 4*, *Figure 1—figure supplement 5*). FARs from three bumble bee species belonging to the FAR-J ortholog group are nearly identical, sharing 97.2–99.7% protein sequence identity; *Bluc*FAR-A1 and *Bter*FAR-A1 share 99.4% protein sequence identity with each other and 60.9–61.1% with *Blap*FAR-A1. *Bluc*FAR-A2 and *Bter*FAR-A2 share 94.8% protein sequence identity (*Figure 1—figure supplement 4*). *Bluc*FAR-J was not cloned because of its very high similarity to *Bter*FAR-J (99.7% sequence identity, two amino acid differences). We cloned two versions of *Blap*FAR-A1: one that was custom-synthesized based on the predicted full-length coding sequence assembled from RNA-Seq data and one called *Blap*FAR-A1-short that we consistently PCR-amplified from *B. lapidarius* male LG cDNA. *Blap*FAR-A1-short has an in-frame internal 66 bp deletion in the coding region that does not disrupt the predicted active site, putative NAD(P)$^+$ binding site or putative substrate binding site (*Figure 6—figure supplement 1C*). Using RT-qPCR with specific primers for each variant, we confirmed that both *Blap*FAR-A1 and *Blap*FAR-A1-short are expressed in the *B. lapidarius* male LG and virgin queen LG (*Figure 4—figure supplement 2*).

To test whether the MMP-biosynthetic FAR candidates code for enzymes with fatty acyl reductase activity and to uncover their substrate specificities, we cloned the candidate FAR coding regions into yeast expression plasmids, heterologously expressed the FARs in *Saccharomyces cerevisiae* and assayed the fatty alcohol production by GC (*Figure 6—figure supplement 2*, *Figure 6—figure supplement 3*).

His-tagged FARs were detected in all yeast strains transformed with plasmids bearing FARs (*Figure 6—figure supplement 4*, *Figure 6—figure supplement 1A*), while no His-tagged proteins were detected in the negative control (yeasts transformed with an empty plasmid). In addition to the major protein bands corresponding to the theoretical FAR molecular weight, we typically observed protein bands with lower and/or higher molecular weight (*Figure 6—figure supplement 4*). The synthetic *Bluc*FAR-A1-opt and *Bluc*FAR-A2-opt coding regions with codon usage optimized for *S. cerevisiae* showed a single major western blot signal corresponding to the position of the predicted full-length protein (*Figure 6—figure supplement 4*). The shortened heterologously expressed proteins thus presumably represent incompletely transcribed versions of full-length FARs resulting from ribosome stalling (*Angov, 2011*), while the higher molecular weight bands might correspond to aggregates of full-length and incompletely translated FARs. Because the codon-optimized *Bluc*FAR-A1-opt and *Bluc*FAR-A2-opt exhibit the same overall specificity in yeast expression system as the respective non-codon-optimized FARs (*Figure 6—figure supplement 5*), we employed non-codon-optimized FARs for further functional characterization. We only used the codon-optimized versions of *Bluc*FAR-A1 and *Bluc*FAR-A2 in experiments with exogenously supplemented substrates to increase the possibility of product detection, as the optimized FARs produce overall higher quantities of fatty alcohols (*Figure 6—figure supplement 5*, *Figure 6—source data 1*).

Characterization of FAR enzymatic activities involved identification of numerous individual FA derivatives, denoted using the length of the carbon chain (e.g. 20: for 20-carbon chain), the number of double bonds (either as the position/configuration of the double bond(s), if known, for example Z9, or by ':X', for example :1 and :2 for monounsaturated and diunsaturated FAs, respectively) and the C1 moiety (COOH for acid, OH for alcohol, Me for methyl ester, CoA for CoA-thioester).

Functional characterization of FARs from *B. terrestris* and *B. lucorum* in yeast indicated that saturated C16 to C26 fatty alcohols are produced by both *Bter*FAR-A1 and *Bluc*FAR-A1 and *Bter*FAR-J enzymes (*Figure 6A*, *Figure 6—figure supplement 3*); *Bter/Bluc*FAR-A1 prefers C22 substrates, whereas *Bter*FAR-J has an optimal substrate preference slightly shifted to C24. Unlike any of the other characterized FARs, *Bter*FAR-A1 and *Bluc*FAR-A1 are also capable of reducing supplemented monounsaturated Z15-20: acyl to the corresponding alcohol (*Figure 6B*, *Figure 6—figure supplement 6C*). Both *Bter*FAR-A2 and *Bluc*FAR-A2 reduce only 16: and 18: acyls (*Figure 6A*, *Figure 6—figure supplement 7*).

Characterization of *B. lapidarius* FARs showed that *Blap*FAR-A1, in contrast to *Bter*FAR-A1 and *Bluc*FAR-A1, produces Z9-16:OH and Z9-18:OH (*Figure 6A*, *Figure 6—figure supplement 2*). *Blap*-FAR-A4 produces 16:OH and Z9-16:OH, together with lower quantities of 14:OH and Z9-18:OH (*Figure 6A*, *Figure 6—figure supplement 8A*). *Blap*FAR-A5 produces 16:OH as a major product and lower amounts of 14:OH, Z9-16:OH, 18:OH and Z9-18:OH (*Figure 6A*, *Figure 6—figure supplement 8A*). In addition, both *Blap*FAR-A1 and *Blap*FAR-A4 are capable of reducing supplemented polyunsaturated fatty acyls (Z9,Z12-18: and Z9,Z12,Z15-18:) to their respective alcohols (*Figure 6B*, *Figure 6—figure supplement 6AB*). Similarly to *Bter*FAR-J, *Blap*FAR-J also reduces saturated C16 to C26 acyls (*Figure 6A*, *Figure 6—figure supplement 3*).

No fatty alcohols were detected in the negative control, that is the yeasts transformed with empty plasmid (*Figure 6—figure supplement 2*, *Figure 6—figure supplement 8B*). The total quantities of fatty alcohols accumulated in FAR-expressing yeasts range from few milligrams to tens of milligrams per liter of culture (*Figure 6—figure supplement 8B*), the exceptions being *Bter*FAR-A2 and *Bluc*-FAR-A2 which both produce sub-milligram quantities of fatty alcohols. We did not detect the formation of fatty aldehydes in any of the yeast cultures (data not shown), confirming that the studied FARs are strictly alcohol-forming fatty acyl-CoA reductases. In contrast to *Blap*FAR-A1, *Blap*FAR-A1-short does not produce detectable amounts of any fatty alcohol (*Figure 6—figure supplement 1B*), suggesting that the missing 22-amino acid region (*Figure 6—figure supplement 1C*) is necessary for the retention of FAR activity.

Overall, the FAR specificities determined in yeast correlate well with the composition of LG fatty alcohols and fatty acyls (*Figure 5*). The exceptions are Z9,Z12-18:OH and Z9,Z12,Z15-18:OH, as none of the studied FARs from *B. lucorum* or *B. terrestris* reduce the corresponding acyls.

## Discussion

Since the first genome-scale surveys of gene families, gene duplications and lineage-specific gene family expansions have been considered major mechanisms of diversification and adaptation in eukaryotes (*Lespinet et al., 2002*) and prokaryotes (*Jordan et al., 2001*). Tracing the evolution of gene families and correlating them with the evolution of phenotypic traits has been facilitated by the growing number of next-generation genomes and transcriptomes from organisms spanning the entire tree of life. However, obtaining experimental evidence of the function of numerous gene family members across multiple species or lineages is laborious. Thus, such data are scarce, and researchers have mostly relied on computational inference of gene function (*Radivojac et al., 2013*). Here, we aimed to combine computational inference with experimental characterization of gene function to understand the evolution of the FAR family, for which we predicted a substantial gene number expansion in our initial transcriptome analysis of the buff-tailed bumble bee *B. terrestris* (*Buček et al., 2016*). We specifically sought to determine whether the FARs that emerged through expansion of the FAR gene family substantially contribute to MMP biosynthesis in bumble bees.

We found that the FAR-A group underwent massive expansion in all analyzed species of bumble bee and stingless bee lineages but not in any other Hymenoptera species, with the exceptions of the ant *Camponotus floridanus* and the mining bee *Andrena vaga* (*Figure 1*, *Figure 2*). The phylogenetic sister group relationship of bumble bees and stingless bees (*Peters et al., 2017*; *Branstetter et al., 2017*) indicates that the FAR duplication process occurred or started in their

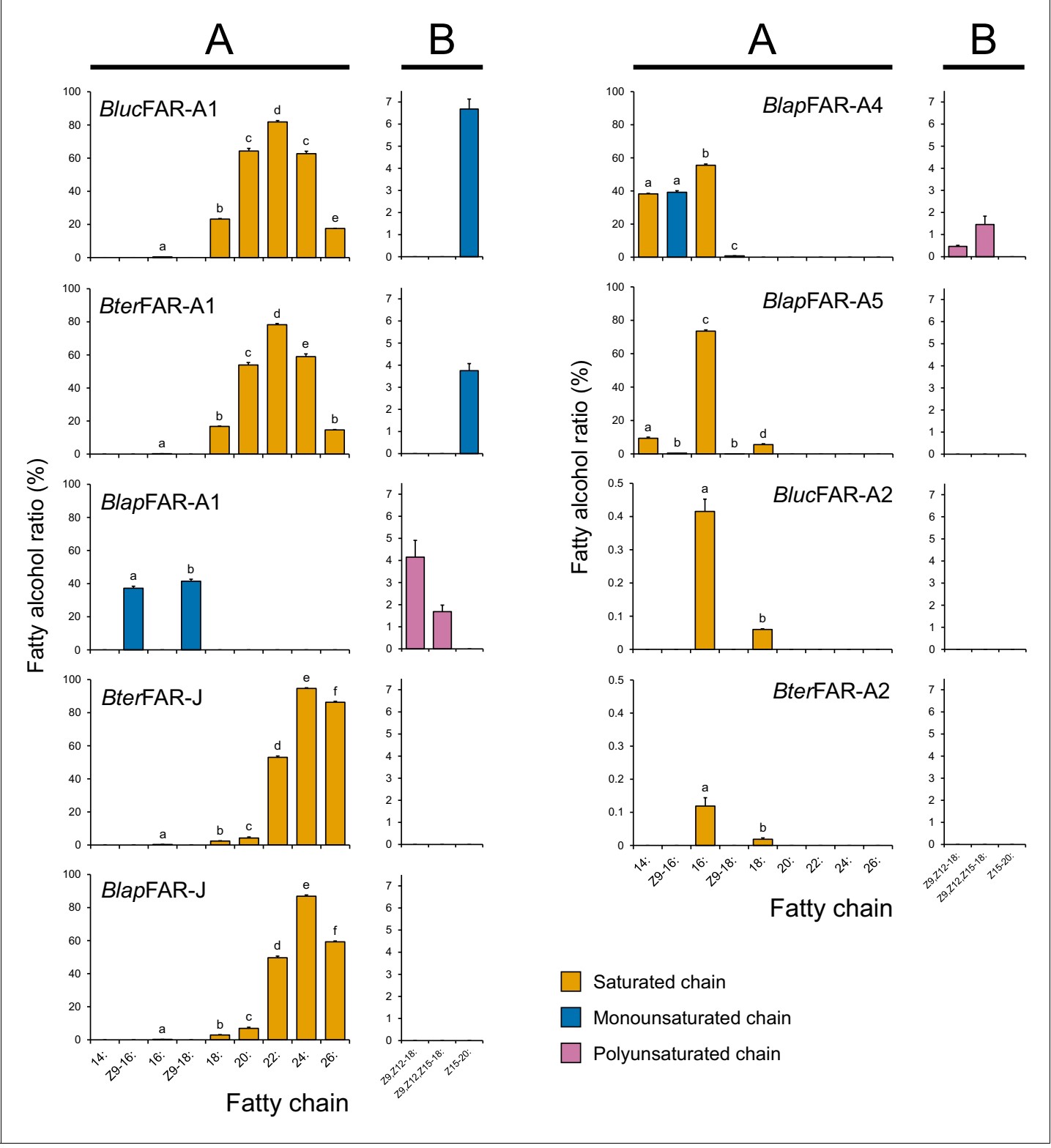

**Figure 6.** Apparent specificity of bumble bee FARs expressed in yeasts. The fatty alcohol ratios represent the apparent specificity of individual MMP-biosynthetic FAR candidates when expressed in yeast hosts (*N* = 3). The fatty alcohol production was quantified by GC analysis of yeast total lipid extracts with the recombinant FARs either acting on yeast native lipids (**A**) or acting on non-native substrates after supplementation of yeasts with either *Z9,Z12-18:*, *Z9,Z12,Z15-18:* or *Z15-20:* acyl-CoA precursors (**B**). The data in (**B**) panel of *Bluc*FAR-A1 and *Bluc*FAR-A2 are taken from yeasts expressing the proteins from yeast codon-optimized nucleotide sequences. Note the different *y*-axis scale for *Bter*FAR-A2 and *Bluc*FAR-A2. Significant differences

*Figure 6 continued on next page*

*Figure 6 continued*

(p < 0.01, one-way ANOVA followed by *post-hoc* Tukey's HSD test) are marked with different letters. See *Equation 1* for a description of fatty alcohol ratio calculation.

DOI: https://doi.org/10.7554/eLife.39231.022

The following source data and figure supplements are available for figure 6:

**Source data 1.** List of fatty alcohol and fatty acyl quantities in FAR-expressing yeasts and fatty alcohol ratios of FARs.

DOI: https://doi.org/10.7554/eLife.39231.031

**Figure supplement 1.** Protein expression and lipid profile in yeast expressing *Blap*FAR-A1-short.

DOI: https://doi.org/10.7554/eLife.39231.023

**Figure supplement 2.** Fatty alcohol production in yeasts expressing bumble bee FARs specific for long chain fatty acyls.

DOI: https://doi.org/10.7554/eLife.39231.024

**Figure supplement 3.** Fatty alcohol production in yeasts expressing bumble bee FARs specific for very long chain fatty acyls.

DOI: https://doi.org/10.7554/eLife.39231.025

**Figure supplement 4.** Protein expression in yeasts expressing bumble bee FARs.

DOI: https://doi.org/10.7554/eLife.39231.026

**Figure supplement 5.** Apparent specificity of FARs expressed from wild-type and yeast codon-optimized sequences.

DOI: https://doi.org/10.7554/eLife.39231.027

**Figure supplement 6.** Fatty alcohol production in FAR-expressing yeasts supplemented with non-native fatty acyl substrates.

DOI: https://doi.org/10.7554/eLife.39231.028

**Figure supplement 7.** Relative amounts of fatty alcohols produced in bumble bee FAR-expressing yeasts.

DOI: https://doi.org/10.7554/eLife.39231.029

**Figure supplement 8.** The quantitative aspects of fatty alcohol production in yeasts expressing bumble bee FARs.

DOI: https://doi.org/10.7554/eLife.39231.030

ancestor, while the increased number of predicted FAR-A genes in *A. vaga* and *C. floridanus* suggests that the FAR-A genes independently underwent duplication in other hymenopteran lineages. According to estimated lineage divergence times, FAR duplication events in bumble bees and stingless bees started after their divergence from *Apis* 54 million years ago (40–69 million years 95% confidence interval) (*Peters et al., 2017*). The number of inferred FAR-A orthologs is inflated by predicted pseudogenes—FARs with fragmented coding sequences that lack some of the catalytically essential domains and motifs (*Figure 1—figure supplement 2*). These predicted catalytically inactive yet highly expressed FAR-A pseudogenes might play a role in regulating the FAR-catalyzed reduction (*Pils and Schultz, 2004*). The number of predicted FAR-A pseudogenes indicate that the FAR-A ortholog group expansion in this lineage was a highly dynamic process (*Figure 1*, *Figure 1—figure supplement 2*). The high number of species-specific FAR-A duplications or losses between the closely related species *B. lucorum* and *B. terrestris*, which diverged approximately 5 million years ago (*Hines, 2008*), further indicates the dynamic evolutionary processes acting on the FAR genes. Notably, gene family expansion inference based on mixed transcriptome and genome data is limited by (1) the uncertainty of distinguishing genuine genes from alternative splice variants and non-overlapping transcript fragments and (2) potential errors in genome assemblies or annotations. The pattern of lineage-specific FAR-A gene expansion in stingless bees and bumble bees is, to the best of our knowledge, corroborated by all genomic and transcriptomic datasets currently available for this group. Confirming many of the other potential lineage-specific FAR gene expansions in Hymenoptera will, however, require analysis of additional genomic resources.

Strikingly, both stingless bees and bumble bees share a life strategy of scent marking using LG secretion (*Jarau et al., 2004*). In worker stingless bees, LG secretion is used as a trail pheromone to recruit nestmates to food resources and generally contains fatty alcohols such as hexanol, octanol, and decanol in the form of their fatty acyl esters (*Jarau et al., 2006*; *Jarau et al., 2010*). The correlation between FAR-A ortholog group expansion and use of LG-produced fatty alcohols as marking pheromones or their precursors suggests a critical role for FAR-A gene group expansion in the evolution of scent marking. In the future, identification and characterization of FAR candidates involved in production of stingless bee worker LG-secretion could corroborate this hypothesis.

Bumble bee orthologs (FAR-G) of *B. mori* pheromone-biosynthetic FAR (*Moto et al., 2003*) are not abundantly or specifically expressed in male bumble bee LGs, as evidenced by RNA-Seq RPKM values (*Figure 1—figure supplement 1*). MMP-biosynthetic FARs in bumble bees and female sex

pheromone-biosynthetic FARs in moths (Lepidoptera) were therefore most likely recruited independently for the tasks of pheromone biosynthesis.

Various models have attempted to describe the evolutionary mechanisms leading to the emergence and maintenance of gene duplicates (*Innan and Kondrashov, 2010*). The fragmented state of the *B. terrestris* genome and its limited synteny with the *A. mellifera* genome restricts our ability to reconstruct the genetic events accompanying the FAR duplications resulting in the FAR-A ortholog group expansion. Notably, the MMP quantities in bumble bees are substantially higher than quantities of pheromones in other insects of comparable size. For example, *B. terrestris*, *B. lucorum* and *B. lapidarius* bumble bee males can produce several milligrams of MMPs (Kindl, personal communication, *Figure 5*), while the sphingid moth *Manduca sexta* produces tens of nanograms of sex pheromone (*Tumlinson et al., 1989*). Taking into consideration the large quantities of MMPs in bumble bee males, we speculate that gene dosage benefits could substantially contribute to the duplications and duplicate fixation of MMP-biosynthetic FARs. Under this model, sexual selection favoring bumble bee males capable of producing large quantities of MMPs could fix the duplicated FARs in a population (*Lespinet et al., 2002*; *Innan and Kondrashov, 2010*).

Mechanistically, gene duplications can be facilitated by associated TEs (*Reams and Roth, 2015*; *Schrader and Schmitz, 2018*). The content of repetitive DNA in the *B. terrestris* and *B. impatiens* genome assemblies is 14.8% and 17.9%, respectively (*Sadd et al., 2015*), which is lower than in other insects such as the beetle *Tribolium castaneum* (30%), *Drosophila* (more than 20%) or the parasitoid wasp *Nasonia vitripenis* (more than 30%), but substantially higher than in the honey bee *Apis mellifera* (9.5%) (*Elsik et al., 2014*; *Weinstock and Honeybee Genome Sequencing Consortium, 2006*). Our finding that TEs are enriched in the vicinity of FAR-A genes in the *B. terrestris* and *B. impatiens* genomes indicates that TEs presumably contributed to the massive expansion of the FAR-A ortholog group (*Figure 3*). Expansions of another pheromone biosynthetic gene family, FADs, in the fly *Drosophila* and in corn borer moths (*Ostrinia*) also have been found to be mediated by TEs. It was proposed that up to seven FAD loci in these species originated by repeated retrotransposition or DNA-mediated transposition (*Fang et al., 2009*; *Xue et al., 2007*).

One notable feature of FAR-A expansion in bumble bees and stingless bees is the extent of gene expansion, often generating 10 or more predicted FAR genes (*Figure 2*). One scenario for FAR-A expansion is that TEs provided substrates for non-homologous recombination that led to the initial duplication (*Reams and Roth, 2015*; *Bourque, 2009*). Redundant copies of the ancestral FAR-A gene that arose as a result of the duplication were not under strong purifying selection and might have tolerated accumulation of TEs in their vicinity. The higher abundance of TEs subsequently made the region vulnerable to structural rearrangements, which facilitated expansion of the FAR-A genes. Janoušek *et al.* presented a model in which TEs facilitate gene family expansions in mammalian genomes (*Janoušek et al., 2013*; *Janoušek et al., 2016*). Their model describes gradual accumulation of TEs along with expansions of gene families, which under some model parameters can lead to a runaway process characterized by rapid and accelerating gene family expansion (*Janoušek et al., 2016*). In bumble bees, this runaway process could have been facilitated by fixation of duplicated FAR-A genes due to sexual selection for increased FAR expression. However, alternative scenarios are possible. For example, initial duplication of the FAR-A ancestral gene might have been unrelated to TEs, and TEs may have accumulated after the initial duplication. Alternatively, translocation of the FAR-A ancestral copy to a TE-rich region in the common ancestor of bumble bees and stingless bees might have facilitated expansion of this ortholog group. Further research is needed to confirm which of these scenarios is most likely.

The spectrum of fatty alcohols in *B. terrestris* and *B. lucorum* male LG differs substantially from that of *B. lapidarius*. In both *B. terrestris* and *B. lucorum*, the male LG extract contains a rich blend of C14–C26 fatty alcohols with zero to three double bonds (*Figure 5*). In *B. lapidarius*, the male LG extract is less diverse and dominated by Z9-16:OH and 16:OH (*Figure 5*, *Figure 5—source data 1*). To uncover how the distinct repertoire of FAR orthologs contributes to the biosynthesis of species-specific MMPs, we functionally characterized LG-expressed FARs. We previously found that the transcript levels of biosynthetic genes generally reflect the biosynthetic pathways most active in bumble bee LG (*Buček et al., 2016*; *Buček et al., 2013*). For further experimental characterization, we therefore selected the FAR-A and FAR-J gene candidates, which exhibited high and preferential expression in male LG (*Figure 1—figure supplement 1*). Notably, the abundant expression of *Blap*FAR-A1 and *Bter*FAR-J in both virgin queen and male LG (*Figure 4—figure supplement 1*) suggests that

these FARs might also have been recruited for production of queen-specific signals (*Amsalem et al., 2014*). We found that the highly similar *Bluc*FAR-A1/*Bter*FAR-A1 and *Blap*FAR-A1 orthologs exhibit distinct substrate preferences for longer fatty acyl chains (C18–C26) and shorter monounsaturated fatty acyl chains (*Z*9-16: and *Z*9-18:), respectively. This substrate preference correlates with the abundance of *Z*9-16:OH in *B. lapidarius* MMP and the almost complete absence of *Z*9-16:OH in *B. lucorum* and *B. terrestris* (*Figure 5—source data 1*). *Blap*FAR-A4 and to some extent *Blap*FAR-A5 likely further contribute to the biosynthesis of *Z*9-16:OH in *B. lapidarius*. The ability of *Bluc*FAR-A1 and *Bter*FAR-A1 (and not of *Blap*FAR-A1) to reduce long monounsaturated fatty acyls (*Z*15-20:) also correlates with the absence of detectable amounts of *Z*15-20:OH in *B. lapidarius* MMP.

Our comprehensive GC analysis of bumble bee male LGs, however, indicates that the composition of LG fatty acyls is another factor that contributes substantially to the final MMP composition. For example, the very low quantities of *Z*9-16:OH in *B. terrestris* and *B. lucorum* and of *Z*15-20:OH in *B. lapidarius* (*Figure 5*) can be ascribed to the absence of a FAR with the corresponding substrate specificity, but the availability of potential substrates, that is very low amount of *Z*9-16: acyl in *B. lucorum* and *B. terrestris* male LG and the absence of detectable *Z*15-20: acyl in *B. lapidarius* LG also likely contribute.

We detected several fatty alcohols in FBs of *B. terrestris* and *B. lucorum*, 16:OH, *Z*9,*Z*12-18:OH and *Z*9,*Z*12,*Z*15-18:OH being the most abundant (*Figure 5—source data 1*, *Figure 5—figure supplement 1*). Fatty alcohols are not expected to be transported from FB across haemolymph to LG (*Buček et al., 2016*). However, the presence of *Z*9,*Z*12-18: and *Z*9,*Z*12,*Z*15-18: fatty alcohols in FB provides an explanation for why we did not find a FAR reducing *Z*9,*Z*12-18: and *Z*9,*Z*12,*Z*15-18: among the functionally characterized candidates from *B. terrestris* and *B. lucorum*. Our candidate selection criteria were based on the LG-specific FAR transcript abundance, and we might have disregarded a FAR that is capable of polyunsaturated fatty acyl reduction and is expressed at comparable levels in both LG and FB.

We noted several discrepancies between the FAR specificity in the yeast expression system and the apparent FAR specificity *in vivo* (i.e. the apparent specificity of fatty acyl reduction in bumble bee LG calculated from the fatty acyl and fatty alcohol content, *Figure 5—figure supplement 1*). We found that *Blap*FAR-A1 is capable of producing substantial amounts of *Z*9-16:OH and *Z*9-18:OH (*Figure 6—figure supplement 8A*) in the yeast system, while in *B. lapidarius* LG, only *Z*9-16: acyl is converted to *Z*9-16:OH, as evidenced by the absence of detectable amounts of *Z*9-18:OH (*Figure 5*). Additionally, *Blap*FAR-A1 and *Blap*FAR-A4 produce in the yeast expression system polyunsaturated fatty alcohols (*Figure 6*) that are not present in *B. lapidarius* male LG, despite the presence of corresponding fatty acyls in the LG (*Figure 5*). A possible explanation for the differences between FAR specificities in the bumble bee LG and the yeast expression system is that the pool of LG fatty acyls that we assessed and used to evaluate the apparent FAR specificities has a different composition than the LG pool of fatty acyl-CoAs, which are the form of fatty acyls accepted by FARs as substrates. The presumably low concentrations of *Z*9,*Z*12-18:CoA, *Z*9,*Z*12,*Z*15-18:CoA, and *Z*9-18:CoA in the LG of male *B. lapidarius* compared to the concentrations of the respective fatty acyls could prevent detectable accumulation of the corresponding fatty alcohols. We therefore propose that the selectivity of enzymes and binding proteins that convert fatty acyls to fatty acyl-CoAs (*Oba et al., 2004*) and protect fatty acyl-CoAs from hydrolysis (*Matsumoto et al., 2001*) represents an additional mechanism shaping the species-specific fatty alcohol composition in bumble bee male LGs.

In sum, the functional characterization of bumble bee FARs indicates that the combined action of FARs from the expanded FAR-A ortholog group has the capability to biosynthesize the majority of bumble bee MMP fatty alcohols. The substrate specificity of FARs apparently contributes to the species-specific MMP composition, but other biosynthetic steps, namely the process of fatty acyl and fatty acyl-CoA accumulation, likely also contribute to the final fatty alcohol composition of bumble bee MMPs.

## Conclusion

In the present work, we substantially broadened our limited knowledge of the function of FARs in Hymenoptera, one of the largest insect orders. The experimentally determined reductase specificity of FARs that are abundantly expressed in bumble bee male LGs is consistent with their role in MMP biosynthesis. The majority of these MMP-biosynthetic FARs belong to the FAR-A ortholog group. Reconstruction of the FAR gene family evolution indicates the onset of FAR-A gene expansion in the

common ancestor of bumble bee and stingless bee lineages after their divergence from honey bee lineage. We therefore propose that the strategy of bumble bees and stingless bees to employ fatty alcohols as marking pheromones was shaped by FAR gene family expansion. Our analysis of TE distribution in the *B. terrestris* genome indicates that TEs enriched in the vicinity of FAR-A genes might have substantially contributed to the dramatic expansion of the FAR-A gene group. In the future, the increasing availability of annotated Hymenopteran genome assemblies should enable us to more precisely delineate the taxonomic extent and evolutionary timing of the massive FAR gene family expansion and assess in detail the role of TEs in the process.

# Materials and methods

## Key resources table

| Reagent type (species) or resource | Designation | Source or reference | Identifiers | Additional information |
|---|---|---|---|---|
| Gene (*Acromyrmex echinatior*) | *Acromyrmex echinatior* genome | https://www.ncbi.nlm.nih.gov/bioproject | PRJNA271903 | |
| Gene (*Andrena vaga*) | *Andrena vaga* transcriptome | https://www.ncbi.nlm.nih.gov/bioproject | PRJNA252325 | |
| Gene (*Apis mellifera*) | *Apis mellifera* genome | https://www.ncbi.nlm.nih.gov/bioproject | PRJNA13343 | |
| Gene (*Bombus impatiens*) | *Bombus impatiens* genome | https://www.ncbi.nlm.nih.gov/bioproject | PRJNA70395 | |
| Gene (*Bombus rupestris*) | *Bombus rupestris* transcriptome | https://www.ncbi.nlm.nih.gov/bioproject | PRJNA252240 | |
| Gene (*Bombus terrestris*) | *Bombus terrestris* genome | https://www.ncbi.nlm.nih.gov/bioproject | PRJNA68545 | |
| Gene (*Bombus terrestris*) | *B. terrestris* LG and FB transcriptomes | https://www.ncbi.nlm.nih.gov/bioproject | PRJEB9937 | |
| Gene (*Camponotus floridanus*) | *Camponotus floridanus* genome | https://www.ncbi.nlm.nih.gov/bioproject | PRJNA50201 | |
| Gene (*Camptopoeum sacrum*) | *Camptopoeum sacrum* transcriptome | https://www.ncbi.nlm.nih.gov/bioproject | PRJNA252153 | |
| Gene (*Ceratina calcarata*) | *Ceratina calcarata* genome | https://www.ncbi.nlm.nih.gov/bioproject | PRJNA340002 | |
| Gene (*Colletes cunicularius*) | *Colletes cunicularius* transcriptome | https://www.ncbi.nlm.nih.gov/bioproject | PRJNA252324 | |
| Gene (*Dufourea novaeangliae*) | *Dufourea novaeangliae* genome | https://www.ncbi.nlm.nih.gov/bioproject | PRJNA311229 | |
| Gene (*Epeolus variegatus*) | *Epeolus variegatus* transcriptome | https://www.ncbi.nlm.nih.gov/bioproject | PRJNA252262 | |
| Gene (*Euglossa dilemma*) | *Euglossa dilemma* transcriptome | https://www.ncbi.nlm.nih.gov/bioproject | PRJNA252310 | |
| Gene (*Harpegnathos saltator*) | *Harpegnathos saltator* genome | https://www.ncbi.nlm.nih.gov/bioproject | PRJNA273397 | |

*Continued on next page*

*Continued*

| Reagent type (species) or resource | Designation | Source or reference | Identifiers | Additional information |
|---|---|---|---|---|
| Gene (*Megachile rotundata*) | *Megachile rotundata* genome | https://www.ncbi.nlm.nih.gov/bioproject | PRJNA87021 | |
| Gene (*Melipona quadrifasciata*) | *Melipona quadrifasciata* genome | https://www.uniprot.org/proteomes/ | UP000053105 | |
| Gene (*Melitta haemorrhoidalis*) | *Melitta haemorrhoidalis* transcriptome | https://www.ncbi.nlm.nih.gov/bioproject | PRJNA252208 | |
| Gene (*Nasonia vitripenis*) | *Nasonia vitripenis* genome | https://www.ncbi.nlm.nih.gov/bioproject | PRJNA20073 | |
| Gene (*Panurgus dentipes*) | *Panurgus dentipes* transcriptome | https://www.ncbi.nlm.nih.gov/bioproject | PRJNA252205 | |
| Gene (*Polistes canadensis*) | *Polistes canadensis* genome | https://www.ncbi.nlm.nih.gov/bioproject | PRJNA301748 | |
| Gene (*Tetragonula carbonaria*) | *Tetragonula carbonaria* transcriptome | https://www.ncbi.nlm.nih.gov/bioproject | PRJNA252285 | |
| Gene | *B. lucorum* and *B. lapidarius* transcriptomes | This paper, https://www.ncbi.nlm.nih.gov/bioproject | PRJNA436452 | |
| Gene (*Bombus lapidarius*) | Cloned CDS of FAR-A1, FAR-A1-short, FAR-A4, FAR-A5, FAR-J | This paper, https://www.ncbi.nlm.nih.gov/genbank/ | MG450698; MG450699; MG450702; MG450703; MG450701 | |
| Gene (*Bombus lucorum*) | Cloned CDS of FAR-A1, FAR-A1-opt, FAR-A2, FAR-A2-opt | This paper, https://www.ncbi.nlm.nih.gov/genbank/ | MG930980; MG450697; MG930982; MG450704 | |
| Gene (*Bombus terrestris*) | Cloned CDS of FAR-A1, FAR-A2, FAR-J | This paper, https://www.ncbi.nlm.nih.gov/genbank/ | MG930981; MG930983; MG450700 | |
| Strain, strain background (*Saccharomyces cerevisiae*) | BY4741 | Brachmann et al., 1998 (DOI: 10.1002/(SICI) 1097-0061 (19980130) 14:2 < 115:: AID-YEA204 > 3.0.CO;2–2) | | *MATa his3Δ1 leu2Δ0 met15Δ0 ura3Δ0* |
| Strain, strain background (*Saccharomyces cerevisiae*, BY4741) | FAR expressing yeast strains | This paper | | See ***Supplementary file 2*** |
| Biological sample (*Bombus lapidarius*) | Labial gland (LG) from males and females; faty body (FB); flight muscle; gut | This paper | | Dissected by standard techniques from individuals reared in laboratory colonies |
| Biological sample (*Bombus lucorum*) | Labial gland (LG) from males and females; faty body (FB); flight muscle; gut | This paper | | Dissected by standard techniques from individuals reared in laboratory colonies |

*Continued on next page*

*Continued*

| Reagent type (species) or resource | Designation | Source or reference | Identifiers | Additional information |
|---|---|---|---|---|
| Biological sample (*Bombus terrestris*) | Labial gland (LG) from males and females; faty body (FB); flight muscle; gut | This paper | | Dissected by standard techniques from individuals reared in laboratory colonies |
| Antibody | Anti-6×His tag antibody-HRP conjugate, mouse monoclonal | Merck | A7058; RRID:AB_258326 | (1:2000) |
| Recombinant DNA reagent | *Escherichia coli* DH5α | Thermo Fisher Scientific | 18265017 | |
| Recombinant DNA reagent | *Escherichia coli* One Shot TOP10 | Thermo Fisher Scientific | C4040 | |
| Recombinant DNA reagent | pYEXTHS-BN (plasmid) | Holz et al., 2002 (DOI: 10.1016/ S1046-5928 (02)00029–3) | | pUC and 2µ origin; LEU2, URA3 and AmpR selectable markers; PCUP1 inducible promoter; N-terminal 6 × His tag and C-terminal Strep II tag |
| Recombinant DNA reagent | pYEXTHS-BN plasmids carrying FAR CDSs | This paper | | See *Supplementary file 2* |
| Sequence-based reagent | Cloning primers | This paper | | See *Supplementary file 2* |
| Sequence-based reagent | RT-qPCR primers | This paper; Horňáková et al., 2010 (DOI: 10.1016/ j.ab.2009.09.019) | | See *Supplementary file 2* |
| Commercial assay or kit | In-Fusion HD Cloning kit | Clontech (Takara) | 639649 | |
| Commercial assay or kit | LightCycler 480 SYBR Green I Master | Roche | 04707516001 | |
| Commercial assay or kit | RNeasy Mini Kit | Qiagen | 74104 | |
| Commercial assay or kit | S.c. EasyComp Transformation Kit | Thermo Fisher Scientific | K505001 | |
| Commercial assay or kit | SMART cDNA Library Construction Kit | Clontech (Takara) | 634901 | |
| Commercial assay or kit | SuperSignal West Femto Maximum Sensitivity Substrate | Thermo Fisher Scientific | 34096 | |
| Commercial assay or kit | TOPO TA Cloning kit | Thermo Fisher Scientific | 450640 | |
| Software, algorithm | Batch conserved domain search | DOI: 10.1093 /nar/gku1221 | | |
| Software, algorithm | BLAST | DOI: 10.1016/S0022 -2836 (05)80360–2 | | |

*Continued on next page*

*Continued*

| Reagent type (species) or resource | Designation | Source or reference | Identifiers | Additional information |
|---|---|---|---|---|
| Software, algorithm | bowtie2 v2.2.6 | DOI: 10.1038/nmeth.1923 | | |
| Software, algorithm | CLC Genomics Workbench software v. 7.0.1 | http://www.clcbio.com | | |
| Software, algorithm | ggtree | DOI: 10.1111/2041-210X.12628 | | |
| Software, algorithm | ht-seq v0.9.1 | DOI: 10.1093/bioinformatics/btu638 | | |
| Software, algorithm | IQTREE v1.5.5 | DOI: 10.1093/molbev/msu300 | | |
| Software, algorithm | mafft v7.305 | DOI: 10.1093/nar/gkf436 | | |
| Software, algorithm | MAUVE 2.4.0 | DOI: 10.1101/gr.2289704 | | |
| Software, algorithm | Primer BLAST | DOI: 10.1186/1471-2105-13-134 | | |
| Software, algorithm | R programming language | R Core Team. R: A language and environment for statistical computing. (2016) | | |

## Insects

Specimens of *Bombus lucorum* and *Bombus lapidarius* were obtained from laboratory colonies established from naturally overwintering bumble bee queens. The *Bombus terrestris* specimens originated from laboratory colonies obtained from a bumble bee rearing facility in Troubsko, Czech Republic.

LG and FB samples used for transcriptome sequencing were prepared from 3-day-old *B. lapidarius* males by pooling tissues from three specimens from the same colony. The cephalic part of the LG and a section of the abdominal peripheral FB were dissected, transferred immediately to TRIzol (Invitrogen), then flash-frozen at −80°C and stored at this temperature prior to RNA isolation.

## RNA isolation and cDNA library construction

For cloning of FARs and RT-qPCR analysis of tissue-specific gene expression, RNA was isolated from individual bumble bee tissues by guanidinium thiocyanate-phenol-chloroform extraction followed by RQ1 DNase (Promega) treatment and RNA purification using the RNeasy Mini Kit (Qiagen). The tissue sample for RNA isolation from virgin queen LGs consisted of pooled glands from two specimens. A nanodrop ND-1000 spectrophotometer (Thermo Fisher) was employed to determine the isolated RNA concentration. The obtained RNA was kept at −80°C until further use.

The cDNA libraries of LGs from 3-day-old bumble bee males were constructed from 0.50 µg total RNA using the SMART cDNA Library Construction Kit (Clontech) with either Superscript III (Invitrogen) or M-MuLV (New England Biolabs) reverse transcriptase.

## Transcriptome sequencing, assembly and annotation

The male LG and FB transcriptomes of *B. lapidarius* were sequenced and assembled as previously described for the transcriptomes of male LGs and FBs of *B. lucorum* and *B. terrestris* (*Buček et al., 2013*; *Prchalová et al., 2016*). Briefly, total RNA was isolated from the LGs and FBs of three 3-day-old *B. lapidarius* males and pooled into a one FB and one LG sample. Total RNA (5 µg) from each of the samples was used as starting material. Random primed cDNA libraries were prepared using poly (A)+ enriched mRNA and standard Illumina TrueSeq protocols (Illumina). The resulting cDNA was fragmented to an average of 150 bp. RNA-Seq was carried out by Fasteris (Fasteris) and was performed using an Illumina HiSeq 2500 Sequencing System. Quality control, including filtering high-

quality reads based on the fastq score and trimming the read lengths, was carried out using CLC Genomics Workbench software v. 7.0.1 (http://www.clcbio.com). The complete transcriptome libraries were assembled *de novo* using CLC Genomics Workbench software. FAR expression values were calculated by mapping Illumina reads against the predicted coding regions of FAR sequences using bowtie2 v2.2.6 (*Langmead and Salzberg, 2012*) and counting the mapped raw reads using ht-seq v0.9.1 (*Anders et al., 2015*). The raw read counts were normalized for the FAR coding region length and the total number of reads in the sequenced library, yielding reads per kilobase of transcript per million mapped reads (RPKM) values (*Mortazavi et al., 2008*). A constant value of 1 was added to each RPKM value and subsequently log2-transformed and visualized as heatmaps using the ggplot2 package in R (*Core Team R, 2016*). Complete short read (Illumina HiSeq2500) data for FB and LG libraries from *B. lapidarius* and previously sequenced *B. lucorum* (*Buček et al., 2013*) were deposited in the Sequence Read Archive (https://www.ncbi.nlm.nih.gov/sra) with BioSample accession numbers SAMN08625119, SAMN08625120, SAMN08625121, and SAMN08625122 under BioProject ID PRJNA436452.

## FAR sequence prediction

The FARs of *B. lucorum* and *B. lapidarius* were predicted based on Blast2GO transcriptome annotation and their high protein sequence similarity to previously characterized FARs from the European honey bee *Apis mellifera* (*Teerawanichpan et al., 2010b*) and the silk moth *Bombyx mori* (*Moto et al., 2003*).

FAR sequences from species across the Hymenoptera phylogeny were retrieved from publicly available resources. When available, genome assembly-derived FAR sequences were used instead of transcriptome assembly-derived sequences to minimize the impact of misidentification of alternative splice variants as distinct genes on inference of FAR gene expansion. However, in the *Euglossa dilemma* genome-derived proteome, we failed to identify a FAR-A gene, but we did detect FAR-As in the transcriptome sequencing-derived dataset. We therefore used the *Euglossa dilemma* transcriptome rather than genome for downstream analyses. FARs from annotated genomes (*Bombus impatiens* (*Sadd et al., 2015*), *Bombus terrestris* (*Sadd et al., 2015*), *Apis mellifera* (*Janoušek et al., 2016*), *Camponotus floridanus* (*Bonasio et al., 2010*), *Acromyrmex echinatior* (*Nygaard et al., 2011*), *Harpegnathos saltator* (*Bonasio et al., 2010*), *Nasonia vitripenis* (*Werren et al., 2010*), *Polistes canadensis* (*Patalano et al., 2015*), *Dufourea novaeangliae* (*Kapheim et al., 2015*), *Ceratina calcarata* (*Rehan et al., 2016*), *Melipona quadrifasciata* (*Woodard et al., 2011*) and *Megachile rotundata* (*Woodard et al., 2011*)) of other hymenopteran species were retrieved by blastp (*Altschul et al., 1990*) searches (*E*-value cutoff $10^{-5}$) of the species-specific NCBI RefSeq protein database or UniProt protein database using predicted protein sequences of *B. lucorum*, *B. lapidarius* and *B. terrestris* FARs (accessed February 2017). An additional round of blastp searches using FARs found in the first blastp search round did not yield any additional significant (*E*-value $<10^{-5}$) blastp hits, indicating that all FAR homologs were found in the first round of blastp searches (data not shown).

FARs from non-annotated transcriptomes (*Bombus rupestris* (*Peters et al., 2017*), *Tetragonula carbonaria* (*Peters et al., 2017*), *Euglossa dilemma* (*Peters et al., 2017*), *Epeolus variegatus* (*Peters et al., 2017*), *Colletes cunicularius* (*Peters et al., 2017*), *Melitta haemorrhoidalis* (*Peters et al., 2017*), *Camptopoeum sacrum* (*Peters et al., 2017*), *Panurgus dentipes* (*Peters et al., 2017*), and *Andrena vaga* (*Peters et al., 2017*)) were retrieved via local tblastn search (*E*-value cutoff $10^{-5}$) of the publicly available contig sequences (BioProjects PRJNA252240, PRJNA252285, PRJNA252310, PRJNA252262, PRJNA252324, PRJNA252208, PRJNA252153, PRJNA252205, and PRJNA252325) using *Bombus* FARs as a query. The longest translated ORFs were used as a query in tblastn searches against NCBI non-redundant nucleotide database (nr/nt) and ORFs not yielding highly scoring blast hits annotated as FARs were rejected. For FARs with multiple splice variants predicted from the genome sequence, only the longest protein was used for gene tree reconstruction. In the case of transcriptome assembly-derived FARs, we predicted as alternative splice variants those FAR transcripts that were truncated but otherwise identical in sequence to another FAR transcript in the transcriptome. These FARs were not included in gene tree reconstruction.

The active site, conserved Rossmann fold NAD(P)$^+$ binding domain (NABD) (*Rossmann et al., 1974*) and a putative substrate binding site in FAR coding sequences were predicted using Batch conserved domain search (*Marchler-Bauer et al., 2015*). The matrix of protein identities was

calculated using Clustal Omega with default parameters (https://www.ebi.ac.uk/Tools/msa/clustalo/ accessed February 2018).

## FAR gene tree reconstruction

The protein sequences of predicted hymenopteran FARs were aligned using mafft v7.305. The unrooted gene tree was inferred in IQTREE v1.5.5 with 1000 ultrafast bootstrap approximation replicates (*Minh et al., 2013*), and with a model of amino acid substitution determined by ModelFinder (*Kalyaanamoorthy et al., 2017*) implemented in IQTREE. The tree was visualized and annotated using the ggtree package (*Yu et al., 2016*) in R programming language.

## Genome alignment and TE-enrichment analysis

The genomes of *A. mellifera* and *B. terrestris* were aligned using MAUVE 2.4.0 (*Darling et al., 2004*). The genomic position of predicted *B. terrestris* FAR genes was visually inspected using the NCBI Graphical sequence viewer (accessed January 2018 at Nucleotide Entrez Database).

TE-enrichment analysis in the vicinity of FAR genes in the *B. terrestris* and *B. impatiens* genomes was carried out to explore the impact of TEs in extensive expansion of FAR-A genes. TE annotation using the NCBI RepeatMasker provided insufficient detail. Thus, TE annotations for *B. terrestris* and *B. impatiens* were obtained from the Human Genome Sequencing Center FTP (ftp://ftp.hgsc.bcm. edu/; accessed October 2018). We used the approach described by Sadd *et al.* to identify different types of TEs (*Sadd et al., 2015*). For *B. terrestris*, genome version 2.1 was used, and for *B. impatiens* version 1.0 was used. TE density around FAR genes was calculated 10 kb upstream and downstream of each FAR gene, separately for FAR-A genes and non-FAR-A genes. Statistical significance was assessed by permutation test. We compared FAR-A/non-FAR-A gene set average TE density to the null distribution of the average TE densities around *B. terrestris* and *B. impatiens* genes built from 10,000 randomly sampled gene sets with size corresponding to that of the FAR-A/non-FAR-A gene set from the publicly available RefSeq gene set downloaded for the respective genome versions from the NCBI FTP. TE densities were analyzed for a pooled set of all TEs and separately for each TE class and major TE family (Class I: LINE, LTR, LARD, DIRS; Class II: DNA, TIR, MITE, TRIM, Maveric, Helitron) using custom shell scripts and bedtools (*Supplementary file 3*), a suite of Unix genomic tools (*Quinlan and Hall, 2010*). R programming language was used for statistical analysis.

## Quantitative analysis of FAR expression

First-strand cDNA was synthesized from 0.30 µg total RNA using oligo(dT)$_{12-18}$ primers and Superscript III reverse transcriptase. The resulting cDNA samples were diluted 5-fold with water prior to RT-qPCR. The primers used for the assay (*Supplementary file 2*) were designed with Primer-BLAST (https://www.ncbi.nlm.nih.gov/tools/primer-blast/) (*Ye et al., 2012*) and tested for amplification efficiency and specificity by employing amplicon melting curve analysis on dilution series of pooled cDNAs from each species.

The reaction mixtures were prepared in a total volume of 20 µL consisting of 2 µL sample and 500 nM of each primer using LightCycler 480 SYBR Green I Master kit (Roche). The reactions were run in technical duplicates for each sample. RT-qPCR was performed on a LightCycler 480 Instrument II (Roche) in 96-well plates under the following conditions: 95°C for 60 s, then 45 cycles of 95°C for 30 s, 55°C for 30 s and 72°C for 30 s followed by a final step at 72°C for 2 min.

The acquired data were processed with LightCycler 480 Software 1.5 (Roche) and further analyzed with MS Excel (Microsoft Corporation). FAR transcript abundances were normalized to the reference genes phospholipase A2 (PLA2) and elongation factor 1α (eEF1α) as described (*Hornáková et al., 2010*).

## FAR gene isolation and cloning

The predicted coding regions of FARs from *B. lucorum*, *B. lapidarius* and *B. terrestris* were amplified by PCR from LG cDNA libraries using gene-specific primers (*Supplementary file 2*) and Phusion HF DNA polymerase (New England Biolabs). Parts of the full-length coding sequence of *Blap*FAR-A5 were obtained by RACE procedure using SMART cDNA Library Construction Kit. The PCR-amplified sequences containing the 5' and 3' ends of *Blap*FAR-A5 were inserted into pCRII-TOPO vector using TOPO TA Cloning kit (Invitrogen) and sequenced by Sanger method. The resulting sequences

overlapped with contig sequences retrieved from the *B. lapidarius* transcriptome. The full-length *Blap*FAR-A5-coding region was subsequently isolated using gene-specific PCR primers. The sequence of *Blap*FAR-A1 and yeast codon-optimized sequences of *Bluc*FAR-A1-opt and *Bluc*FAR-A2-opt were obtained by custom gene synthesis (GenScript); see *Supplementary file 2* for synthetic sequences. The individual FAR coding regions were then inserted into linearized pYEXTHS-BN vector (*Holz et al., 2002*) using the following restriction sites: *Bter/Bluc*FAR-A1 and *Blap*FAR-J at *Sph*I-*Not*I sites; *Bter/Bluc*FAR2, *Blap*FAR-A1, *Blap*FAR-A1-short and *Blap*FAR-A5 at *Bam*HI-*Not*I sites; and *Bluc*FAR-A1-opt/FAR-A2-opt and *Blap*FAR-A4 at *Bam*HI-*Eco*RI sites. In the case of *Bter*FAR-J, the *Taq* DNA polymerase (New England Biolabs)-amplified sequence was first inserted into pCRII-TOPO vector and then subcloned into pYEXTHS-BN via *Bam*HI-*Eco*RI sites using the In-Fusion HD Cloning kit (Clontech).

The resulting vectors containing FAR sequences *N*-terminally fused with 6×His tag were subsequently transformed into *E. coli* DH5α cells (Invitrogen). The plasmids were isolated from bacteria with Zyppy Plasmid Miniprep kit (Zymo Research) and Sanger sequenced prior to transformation into yeast. The protein-coding sequences of all studied FARs were deposited to GenBank (see Key Resources Table for accession numbers).

## Functional assay of FARs in yeast

Expression vectors carrying FAR-coding sequences were transformed into *Saccharomyces cerevisiae* strain BY4741 (*MAT*a *his3Δ1 leu2Δ0 met15Δ0 ura3Δ0*) (*Brachmann et al., 1998*) using S.c. EasyComp Transformation Kit (Invitrogen). To test FAR specificity, yeasts were cultured for 3 days in 20 mL synthetic complete medium lacking uracil (SC−U) supplemented with 0.5 mM Cu$^{2+}$ (inducer of heterologous gene expression), 0.2% peptone and 0.1% yeast extract. The yeast cultures were then washed with water and the cell pellets lyophilized before proceeding with lipid extraction. FAR specificities were determined with the FARs acting on natural substrates present in yeast cells and with individual fatty acyls added to the cultivation media, with the respective fatty alcohols present in the LGs of studied bumble bees. For this purpose, yeast cultures were supplemented with the following fatty acyls: 0.1 mM *Z*9,*Z*12-18:COOH (linoleic acid, Sigma-Aldrich), *Z*9,*Z*12,*Z*15-18:COOH (α-linolenic acid, Sigma-Aldrich) or *Z*15-20:Me solubilized with 0.05% tergitol. We chose *Z*15-20: as a representative monounsaturated >C20 fatty acyl substrate because *Z*15-20:OH is the most abundant monounsaturated fatty alcohol in *B. terrestris* LG (*Figure 5*).

The level of heterologous expression of bumble bee FARs was assayed by western blot analysis of the whole-cell extracts (obtained via sonication) using anti-6×His tag antibody-HRP conjugate (Sigma-Aldrich) and SuperSignal West Femto Maximum Sensitivity Substrate kit (Thermo Fisher Scientific).

## Lipid extraction and transesterification

Lipids were extracted from bumble bee tissue samples under vigorous shaking using a 1:1 mixture of $CH_2Cl_2$/MeOH, followed by addition of an equal amount of hexane and sonication. The extracts were kept at −20°C prior to GC analysis.

Base-catalyzed transesterification was performed as described previously (*Matousková et al., 2008*) with modifications: the sample was shaken vigorously with 1.2 mL $CH_2Cl_2$/MeOH 2:1 and glass beads (0.5 mm) for 1 hr. After brief centrifugation to remove particulate debris, 1 mL supernatant was evaporated under nitrogen, and the residue was dissolved using 0.2 mL 0.5 M KOH in methanol. The mixture was shaken for 0.5 hr and then neutralized by adding 0.2 mL solution of $Na_2HPO_4$ and $KH_2PO_4$ (0.25 M each) and 35 μL 4 M HCl. The obtained FAMEs were extracted with 600 μL hexane and analyzed by gas chromatography.

For quantification purposes, either 1-bromodecane (10:Br) or 1-bromoeicosane (20:Br) were added to the extracts as internal standards.

## Gas chromatography and fatty alcohol ratio determination

Standards of *Z*9,*Z*12,*Z*15-18:OH and *Z*15-20:OH were prepared from their corresponding acids/FAMEs by reduction with $LiAlH_4$. The *Z*9-18:Me standard was prepared by reacting oleoyl chloride with methanol. Other FAME and fatty alcohol standards were obtained from Nu-Chek Prep and Sigma-Aldrich. The FA-derived compounds in extracts were identified based on the comparison of

their retention times with the standards and comparison of measured MS spectra with those from spectral libraries. Double bond positions were assigned after derivatization with dimethyl disulfide (*Carlson et al., 1989*).

The fatty alcohol ratio is calculated according to *Equation 1*,

$$\text{Fatty alcohol ratio} = \frac{n(\text{Fatty alcohol X})}{n(\text{Fatty alcohol X}) + n(\text{Fatty acyl X})} 100\% \tag{1}$$

where $n$ is the amount in moles and X is the fatty chain structure of certain length, degree of unsaturation and double bond position/configuration. The fatty acyl term in *Equation 1* stands for all transesterifiable fatty acyls present in the sample, for example free FAs, fatty acyl-CoAs, and triacylglycerols, containing the same fatty chain structure. The fatty alcohol ratio thus represents the hypothetical degree of conversion of total fatty acyls (as if they were available as FAR substrates, that is fatty acyl-CoAs) to the respective fatty alcohol and reflects the apparent FAR specificity in the investigated bumble bee tissue or yeast cell.

## GC-FID

GC with flame-ionization detector (FID) was used for quantitative assessment of the FA-derived compounds. The separations were performed on a Zebron ZB-5ms column (30 m × 250 μm I. D. × 0.25 μm film thickness, Phenomenex) using a 6890 gas chromatograph (Agilent Technologies) with following parameters: helium carrier gas, 250°C injector temperature, and 1 mL.min$^{-1}$ column flow. The following oven temperature program was used: 100°C (held for 1 min), ramp to 285°C at a rate of 4 °C.min$^{-1}$ and a second ramp to 320°C at a rate of 20 °C.min$^{-1}$ with a final hold for 5 min at 320°C. The analytes were detected in FID at 300°C using a makeup flow of 25 mL.min$^{-1}$ (nitrogen), hydrogen flow of 40 mL.min$^{-1}$, air flow of 400 mL.min$^{-1}$ and acquisition rate of 5 Hz. The collected data were processed in Clarity (DataApex).

## Comprehensive gas chromatography-mass spectrometry (GC×GC-MS)

The technique was used for initial identification of analytes by comparing their retention characteristics and mass spectra with those of synthetic standards and for quantification of the FA-derived compounds. The following conditions were employed using a 6890N gas chromatograph (Agilent Technologies) coupled to a Pegasus IV D time-of-flight (TOF) mass selective detector (LECO Corp.): helium carrier gas, 250°C injector temperature, 1 mL.min$^{-1}$ column flow, modulation time of 4 s (hot pulse time 0.8 s, cool time 1.2 s), modulator temperature offset of +20°C (relative to secondary oven) and secondary oven temperature offset of +10°C (relative to primary oven). Zebron ZB-5ms (30 m × 250 μm I. D. × 0.25 μm film thickness, Phenomenex) was used as a non-polar primary column and BPX-50 (1.5 m × 100 μm I. D. × 0.10 μm film thickness, SGE) was used as a more polar secondary column. The primary oven temperature program was as follows: 100°C (1 min), then a single ramp to 320°C at a rate of 4 °C.min$^{-1}$ with a final hold for 5 min at 320°C.

The mass selective detector was operated in electron ionization mode (electron voltage −70 V) with a transfer line temperature of 260°C, ion source temperature of 220°C, 100 Hz acquisition rate, mass scan range of 30–600 u and 1800 V detector voltage. ChromaTOF software (LECO Corp.) was used to collect and analyze the data.

## Synthesis of methyl *Z15*-eicosenoate

The *Z15*-20:CoA precursor, methyl *Z15*-eicosenoate (*Z15*-20:Me, **4**), was synthesized by a new and efficient four-step procedure, starting from inexpensive and easily available cyclopentadecanone. The C1–C15 part of the molecule was obtained by Baeyer-Villiger oxidation of cyclopentadecanone, followed by subsequent methanolysis of the resulting lactone **1** and Swern oxidation of the terminal alcohol group of **2**; the C16–C20 fragment was then connected to the aldehyde **3** by Wittig olefination.

All reactions were conducted in flame- or oven-dried glassware under an atmosphere of dry nitrogen. THF, CH$_2$Cl$_2$ and MeOH were dried following standard methods under a nitrogen or argon atmosphere. Petroleum ether (PE, 40–65°C boiling range) was used for chromatographic separations. TLC plates (silica gel with fluorescent indicator 254 nm, Fluka or Macherey-Nagel) were used

for reaction monitoring. Flash column chromatographic separations were performed on silica gel 60 (230–400 mesh, Merck or Acros).

IR spectra were taken on an ALPHA spectrometer (Bruker) as neat samples using an ATR device. $^1$H and $^{13}$C NMR spectra were recorded in CDCl$_3$ on an AV III 400 HD spectrometer (Bruker) equipped with a cryo-probe or an AV III 400 spectrometer (Bruker) equipped with an inverse broad-band probe at 400 MHz for $^1$H and 100 MHz for $^{13}$C. $^1$H NMR chemical shifts were provided in ppm using TMS as external standard; $^{13}$C NMR chemical shifts were referenced against the residual solvent peak. The connectivity was determined by $^1$H-$^1$H COSY experiments. GC-MS (EI) measurements were performed on an Agilent 5975B MSD coupled to a 6890N gas chromatograph (Agilent Technologies). High-resolution MS (HRMS) spectra were measured on a Q-Tof micro spectrometer (resolution 100000 (ESI), Waters) or GCT Premier orthogonal acceleration TOF mass spectrometer (EI and CI, Waters).

## 1-Oxacyclohexadecan-2-one (1)

Cyclopentadecanone (500 mg, 2.23 mmol) was dissolved in dry CH$_2$Cl$_2$ (6 mL) and *meta*-chloroperbenzoic acid (*m*CPBA) (687 mg, 2.79 mmol, 70%) was added at 0°C. The reaction mixture was stirred at room temperature (r.t.), occasionally concentrated under a flow of nitrogen, and the solid residue was re-dissolved in dry CH$_2$Cl$_2$. After stirring for four days, the conversion was still not complete; additional *m*CPBA (164 mg, 667 µmol, 70%) was added at 0°C and stirring was continued at r.t. for 48 hr. The mixture was diluted with CH$_2$Cl$_2$ (20 mL), and the organic layer was washed with saturated NaHCO$_3$ solution (5 × 5 mL) and brine (5 mL). The organic layer was dried over Na$_2$SO$_4$, filtered and evaporated. The crude product was purified by column chromatography (50 mL silica gel, PE/CH$_2$Cl$_2$ 1:1) providing product **1** (426 mg, 80%) as a colorless waxy solid.

**1**: Melting point (m.p.) <30°C. $R_f$ (PE/Et$_2$O 95:5) = 0.5. IR (film): $\nu$ = 2925, 2855, 1733, 1459, 1385, 1349, 1234, 1165, 1108, 1070, 1013, 963, 801, 720 cm$^{-1}$. HRMS (+EI TOF) *m/z*: (C$_{15}$H$_{28}$O$_2$) calc.: 240.2089, found: 240.2090. $^1$H NMR (400 MHz, CDCl$_3$) $\delta$ = 4.13 (t, *J* = 5.7 Hz, 2H, H16), 2.33 (t, *J* = 7.0 Hz, 2H, H3), 1.72–1.56 (m, 4H, H4, H15), 1.48–1.37 (m, 2H, H14), 1.36–1.23 (m, 18H, H5–H13). $^{13}$C NMR (100 MHz, CDCl$_3$) $\delta$ = 174.2, 64.1, 34.6, 28.5, 27.9, 27.28, 27.26, 27.1, 26.8, 26.5, 26.2, 26.1, 26.0, 25.3, 25.1.

## Methyl 15-hydroxypentadecanoate (2)

MeOK (74 µL, 208 µmol, 2.81M in MeOH) was added dropwise at 0°C to a mixture of lactone **1** (50 mg, 208 µmol), dry THF (0.5 mL) and MeOH (1 mL). The mixture was stirred at r.t. for 48 hr, by which point the reaction was complete as indicated by TLC. The solution was quenched with a few drops of water and diluted with Et$_2$O (5 mL). After stirring for 30 min, the layers were separated and the aqueous layer was extracted with Et$_2$O (3 × 3 mL). The combined organic layers were washed with brine and water, dried over Na$_2$SO$_4$, filtered and evaporated to obtain nearly pure product. Purification by column chromatography (5 mL silica gel, PE/EtOAc 9:1) provided product **2** (55 mg, 97%) as a colorless solid.

**2**: m.p. 47–48°C. $R_f$ (PE/Et$_2$O 95:5) = 0.2. IR (film): $\nu$ = 3285, 2917, 2849, 1740, 1473, 1463, 1435, 1412, 1382, 1313, 1286, 1264, 1240, 1217, 1196, 1175, 1117, 1071, 1061, 1049, 1025, 1013, 992, 973, 926, 884, 731, 720, 701 cm$^{-1}$. HRMS (+ESI) *m/z*: (C$_{16}$H$_{32}$O$_3$Na) calc.: 295.2244, found: 295.2245. $^1$H NMR (400 MHz, CDCl$_3$) $\delta$ = 3.64 (s, 3H, OCH$_3$), 3.61 (t, *J* = 6.7 Hz, 2H, H15), 2.28 (t, *J* = 7.5 Hz, 2H, H2), 1.72 (s, 1H, OH), 1.59 (quint, *J* = 7.1 Hz, 2H, H3), 1.54 (quint, *J* = 7.1 Hz, 2H, H14), 1.37–1.14 (m, 20H, H4–H13). $^{13}$C NMR (101 MHz, CDCl$_3$) $\delta$ = 174.5, 63.1, 51.6, 34.2, 32.9, 29.71 (3C), 29.68 (2C), 29.5 (2C), 29.4, 29.3, 25.9, 25.1.

## Methyl 15-oxopentadecanoate (3)

Dry DMSO (110 µL, 1.54 mmol) was added at −78°C dropwise to a mixture of oxalyl chloride (90 µL, 1.03 mmol) and CH$_2$Cl$_2$ (2 mL) in a 25 mL flask, and the reaction mixture was stirred for 15 min. The hydroxy ester **2** (140 mg, 0.51 mmol) in dry CH$_2$Cl$_2$ (2 mL) was added dropwise via a cannula; the white, turbid reaction mixture was stirred for 40 min, and dry triethylamine (432 µL, 3.08 mmol) was added dropwise. The mixture was stirred at −78°C for 1 hr and warmed to 0°C over 30 min at which point the reaction was complete according to TLC. The reaction mixture was diluted with CH$_2$Cl$_2$ (10 mL), quenched with saturated NH$_4$Cl solution (5 mL) and water (5 mL), and warmed to r.t. The layers

were separated and the aqueous layer was extracted with $CH_2Cl_2$ (3 × 10 mL). The combined organic layers were washed with brine, dried over $MgSO_4$, filtered and evaporated. The crude product was purified by flash chromatography (10 mL silica gel, PE/EtOAc 95:5) giving aldehyde **3** (109 mg, 78%) as a colorless waxy solid.

**3**: m.p. <37°C. $R_f$ (PE/EtOAc 9:1) = 0.4. IR (film): $\nu$ = 2923, 2852, 2752, 1738, 1465, 1436, 1362, 1315, 1243, 1197, 1172, 1120, 1017, 985, 958, 883, 811, 719 cm$^{-1}$. HRMS (+CI TOF) $m/z$: ($C_{16}H_{31}O_3$) calc.: 271.2273, found: 271.2277. $^1$H NMR (400 MHz, CDCl$_3$) $\delta$ = 9.76 (t, J = 1.9 Hz, 1H, H15), 3.66 (s, 3H, OCH$_3$), 2.41 (td, J = 7.4, 1.9 Hz, 2H, H14), 2.29 (t, J = 7.5 Hz, 2H, H2), 1.67–1.56 (m, 4H, H3,H13), 1.35–1.20 (m, 18H, H4-H12). $^{13}$C NMR (100 MHz, CDCl$_3$) $\delta$ = 203.1, 174.5, 51.6, 44.1, 34.3, 29.72, 29.70 (2C), 29.58, 29.56, 29.5, 29.4, 29.31, 29.29, 25.1, 22.2.

### Methyl Z15-eicosenoate (4, Z15-20:Me)

NaHMDS (614 μL, 0.614 mmol, 1.0 M in THF) was added dropwise at −55°C over 10 min to a suspension of high vacuum-dried (pentyl)triphenylphosphonium bromide (282 mg, 0.68 mmol) (**Prasad et al., 2014**) in dry THF (3 mL) in a flame dried round-bottomed Schlenk flask. The bright orange reaction mixture was stirred while warming to −40°C for 50 min, and a solution of aldehyde **3** (92 mg, 0.34 mmol) in dry THF (1.5 mL) was added dropwise via cannula at −45°C. Stirring was continued for 1 hr, and the reaction mixture was warmed to r.t. over 90 min. The reaction mixture was diluted with PE (25 mL); filtered through a short silica gel plug, which was washed with PE; and evaporated. The crude product was purified by flash chromatography (silica gel, gradient PE/EtOAc 100:0 to 95:5) to give methyl ester **4** (88 mg, 79%) as a colorless oil.

**4**: $R_f$ (PE/Et$_2$O 95:5) = 0.6. IR (film): $\nu$ = 3005, 2922, 2853, 1743, 1699, 1684, 1653, 1541, 1521, 1507, 1489, 1436, 1362, 1196, 1169, 1106, 1017, 880, 722 cm$^{-1}$. GC-MS (EI) $t_R$ [60°C (4 min) → 10 °C/min to 320°C (10 min)] 21 min; $m/z$ (%): 324 (4) [M$^+$], 292 (26), 250 (10), 208 (9), 152 (7), 123 (12), 111 (22), 97 (48), 87 (40), 83 (52), 74 (56), 69 (70), 59 (14), 55 (100), 41 (43), 28 (26). HRMS (+EI TOF) $m/z$: ($C_{21}H_{40}O_2$) calc.: 324.3028, found: 324.3026. $^1$H NMR (401 MHz, CDCl$_3$) $\delta$ = 5.39–5.30 (m, 2H, H15,H16), 3.66 (s, 3H, OCH$_3$), 2.30 (t, J = 7.5 Hz, 2H, H2), 2.07–1.96 (m, 4H, H14,H17), 1.61 (quint, J = 7.5 Hz, 2H, H3), 1.37–1.14 (m, 24H, H4-H13,H18,H19), 0.89 (t, J = 7.2 Hz, 3H, H20). $^{13}$C NMR (100 MHz, CDCl$_3$) $\delta$ = 174.5, 130.1, 130.0, 51.6, 34.3, 32.1, 29.9, 29.81, 29.79, 29.74, 29.70, 29.68, 29.6, 29.5, 29.4, 29.3, 27.4, 27.1, 25.1, 22.5, 14.2.

### Statistical analysis

All lipid quantifications in yeast and in bumble bee LGs and FBs, and RT-qPCR transcript quantifications in bumble bee tissues were performed using three biological replicates (in addition, technical duplicates were used for RT-qPCR); the number of biological replicates is indicated as N in figures and tables. The results are reported as mean value ±S.D. Significant differences were determined by one-way analysis of variance (ANOVA) followed by post-hoc Tukey's honestly significant difference (HSD) test or by a two-tailed $t$-test as indicated in figures and in Results section.

### Acknowledgements

This work was supported by the Czech Science Foundation (grant no. 15–06569S) and by the Ministry of Education of the Czech Republic (project n. LO1302). Computational resources were provided by the CESNET LM2015042 and the CERIT Scientific Cloud LM2015085, provided under the program 'Projects of Large Research, Development, and Innovations Infrastructures' and by the Okinawa Institute of Science and Technology. MT was supported by Charles University in Prague (project n. SVV260427/2018). We thank Dr. Stefan Jarau for stimulating discussions. We also thank Prof. Baldwyn Torto and two anonymous reviewers for their constructive comments and suggestions.

## Additional information

### Funding

| Funder | Grant reference number | Author |
|---|---|---|
| Czech Science Foundation | 15-06569S | Michal Tupec<br>Aleš Buček<br>Jiří Kindl<br>Irena Valterová |
| Ministerstvo skolstvi, mladeze a telovychovy CR | NPU LO 1302 | Michal Tupec<br>Aleš Buček<br>Petra Wenzelová<br>Iva Pichová |

The funders had no role in study design, data collection and interpretation, or the decision to submit the work for publication.

### Author contributions

Michal Tupec, Validation, Investigation, Visualization, Writing—original draft; Aleš Buček, Conceptualization, Validation, Investigation, Visualization, Methodology, Writing—original draft; Václav Janoušek, Validation, Investigation, Methodology, Writing—original draft; Heiko Vogel, Formal analysis, Validation, Investigation; Darina Prchalová, Jiří Kindl, Investigation, Methodology; Tereza Pavlíčková, Formal analysis, Investigation, Methodology; Petra Wenzelová, Formal analysis, Supervision, Investigation, Writing—original draft; Ullrich Jahn, Data curation, Supervision, Validation, Methodology, Writing—original draft; Irena Valterová, Conceptualization, Data curation, Supervision, Funding acquisition, Validation, Methodology, Writing—original draft; Iva Pichová, Conceptualization, Supervision, Funding acquisition, Validation, Methodology, Writing—original draft

### Author ORCIDs

Michal Tupec (iD) http://orcid.org/0000-0003-2371-4850
Václav Janoušek (iD) http://orcid.org/0000-0002-6894-6826
Heiko Vogel (iD) http://orcid.org/0000-0001-9821-7731
Iva Pichová (iD) https://orcid.org/0000-0002-2178-9819

### Decision letter and Author response

Decision letter https://doi.org/10.7554/eLife.39231.063
Author response https://doi.org/10.7554/eLife.39231.064

## Additional files

### Supplementary files

• Supplementary file 1. Genomic linkage group or scaffold and ortholog group of *B. terrestris* and *A. mellifera* FARs. *B. terrestris* FARs functionally characterized in this study and *A. mellifera* FAR previously functionally characterized are highlighted in green; FAR-A gene orthologs are highlighted in orange; genomic scaffolds not placed into linkage groups are grey.
DOI: https://doi.org/10.7554/eLife.39231.032

• Supplementary file 2. List of primers and synthetic genes and of generated plasmids and strains.
DOI: https://doi.org/10.7554/eLife.39231.033

• Supplementary file 3. Scripts for the analysis of repeat content around FAR genes in bumble bee genomes, statistical analysis and graphics generation.
DOI: https://doi.org/10.7554/eLife.39231.034

• Transparent reporting form
DOI: https://doi.org/10.7554/eLife.39231.035

## Data availability

Complete short read (Illumina HiSeq2500) data from B. lucorum and B. lapidarius were deposited in the Sequence Read Archive (https://www.ncbi.nlm.nih.gov/sra) with BioSample accession numbers SAMN08625119, SAMN08625120, SAMN08625121, and SAMN08625122 under BioProject ID PRJNA436452.

The following dataset was generated:

| Author(s) | Year | Dataset title | Dataset URL | Database and Identifier |
|---|---|---|---|---|
| Tupec M, Buček A | 2018 | Complete short read (Illumina HiSeq2500) data from B. lucorum and B. lapidarius | https://www.ncbi.nlm.nih.gov/bioproject/PRJNA436452 | BioProject, PRJNA436452 |

The following previously published datasets were used:

| Author(s) | Year | Dataset title | Dataset URL | Database and Identifier |
|---|---|---|---|---|
| Peters RS, Krogmann L, Mayer C, Donath A, Gunkel S, Meusemann K, Kozlov A, Podsiadlowski L, Petersen M, Lanfear R, Diez PA, Heraty J, Kjer KM, Klopfstein S, Meier R, Polidori C, Schmitt T, Liu S, Zhou X, Wappler T, Rust J, Misof B, Niehuis O | 2017 | Tetragonula carbonaria transcriptomes | https://www.ncbi.nlm.nih.gov/bioproject/?term=PRJNA252240 | BioProject, PRJNA252240 |
| Sadd BM, Barribeau SM, Bloch G, de Graaf DC, Dearden P, Elsik CG, Gadau J, Grimmelikhuijzen CJ, Hasselmann M, Lozier JD, Robertson HM, Smagghe G, Stolle E, Van Vaerenbergh M, Waterhouse RM, Bornberg-Bauer E, Klasberg S, Bennett AK, Câmara F, Guigó R, Hoff K, Mariotti M, Munoz-Torres M, Murphy T, Santesmasses D, Amdam GV, Beckers M, Beye M, Biewer M, Bitondi MM, Blaxter ML, Bourke AF, Brown MJ, Buechel SD, Cameron R, Cappelle K, Carolan JC, Christiaens O, Ciborowski KL, Clarke DF, Colgan TJ, Collins DH, Cridge AG, Dalmay T, Dreier S, du Plessis L, Duncan E, Erler S, Evans J, Falcon T, Flores K, Freitas FC, Fuchikawa T, Gempe T, Hart- | 2015 | Bombus terrestris genome | https://www.ncbi.nlm.nih.gov/bioproject/?term=PRJNA45869 | BioProject, PRJNA45869 |

felder K, Hauser F

| | | | | |
|---|---|---|---|---|
| Honeybee Genome Sequencing Consortium | 2006 | Apis mellifera genome | https://www.ncbi.nlm.nih.gov/bioproject/?term=PRJNA13343 | BioProject, PRJNA13343 |
| Honeybee Genome Sequencing Consortium | 2010 | Camponotus floridanus genome | https://www.ncbi.nlm.nih.gov/bioproject/?term=PRJNA274144 | BioProject, PRJNA274144 |
| Nygaard S, Zhang G, Schiøtt M, Li C, Wurm Y, Hu H, Zhou J, Ji L, Qiu F, Rasmussen M, Pan H, Hauser F, Krogh A, Grimmelikhuijzen CJ, Wang J, Boomsma JJ | 2011 | Acromyrmex echinatior genome | https://www.ncbi.nlm.nih.gov/bioproject/?term=PRJNA271903 | BioProject, PRJNA271903 |
| Werren JH, Richards S, Desjardins CA, Niehuis O, Gadau J, Colbourne JK; Nasonia Genome Working Group, Werren JH, Colbourne JK | 2011 | Nasonia vitripennis genome | https://www.ncbi.nlm.nih.gov/bioproject/?term=PRJNA20073 | BioProject, PRJNA200 73 |
| Patalano S, Vlasova A, Wyatt C, Ewels P, Camara F, Ferreira PG, Asher CL, Jurkowski TP, Segonds-Pichon A, Bachman M, González-Navarrete I, Minoche AE, Krueger F, Lowy E, Marcet-Houben M, Rodriguez-Ales JL, Nascimento FS, Balasubramanian S, Gabaldon T, Tarver JE, Andrews S, Himmelbauer H, Hughes WO, Guigó R, Reik W, Sumner S | 2015 | Polistes canadensis genome | https://www.ncbi.nlm.nih.gov/bioproject/?term=PRJNA301748 | BioProject, PRJNA30 1748 |
| Woodard SH, Fischman BJ, Venkat A, Hudson ME, Varala K, Cameron SA, Clark AG, Robinson GE | 2011 | Melipona quadrifasciatatranscriptome | https://www.ncbi.nlm.nih.gov/bioproject/?term=PRJNA62691 | BioProject, PRJNA62691 |
| Peters RS, Krogmann L, Mayer C, Donath A, Gunkel S, Meusemann K, Kozlov A, Podsiadlowski L, Petersen M, Lanfear R, Diez PA, Heraty J, Kjer KM, Klopfstein S, Meier R, Polidori C, Schmitt T, Liu S, Zhou X, Wappler T, Rust J, Misof B, Niehuis O | 2017 | Bombus rupestris transcriptomes | https://www.ncbi.nlm.nih.gov/bioproject/?term=PRJNA252285 | BioProject, PRJNA252285 |
| Sadd BM, Barribeau SM, Bloch G, de Graaf DC, Dearden P, Elsik CG, Gadau J, | 2015 | Bombus impatiens genome | https://www.ncbi.nlm.nih.gov/bioproject/?term=PRJNA61101 | BioProject, PRJNA61101 |

| | | | | |
|---|---|---|---|---|
| Grimmelikhuijzen CJ, Hasselmann M, Lozier JD, Robertson HM, Smagghe G, Stolle E, Van Vaerenbergh M, Waterhouse RM, Bornberg-Bauer E, Klasberg S, Bennett AK, Câmara F, Guigó R, Hoff K, Mariotti M, Munoz-Torres M, Murphy T, Santesmasses D, Amdam GV, Beckers M, Beye M, Biewer M, Bitondi MM, Blaxter ML, Bourke AF, Brown MJ, Buechel SD, Cameron R, Cappelle K, Carolan JC, Christiaens O, Ciborowski KL, Clarke DF | | | | |
| Bonasio R, Zhang G, Ye C, Mutti NS, Fang X, Qin N, Donahue G, Yang P, Li Q, Li C, Zhang P, Huang Z, Berger SL, Reinberg D, Wang J, Liebig J | 2010 | Harpegnathos saltator genomes | https://www.ncbi.nlm.nih.gov/bioproject/?term=PRJNA273397 | BioProject, PRJNA273397 |
| Woodard SH, Fischman BJ, Venkat A, Hudson ME, Varala K, Cameron SA, Clark AG, Robinson GE | 2011 | Megachile rotundata transcriptome | https://www.ncbi.nlm.nih.gov/bioproject/?term=PRJNA87021 | BioProject, PRJNA87021 |

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
