## [Decision Letter]

Thank you for submitting your article "Expansion of the fatty acyl reductase gene family shaped pheromone communication in Hymenoptera" for consideration by *eLife*. Your article has been reviewed by three peer reviewers, and the evaluation has been overseen by a guest Reviewing Editor and Ian Baldwin as the Senior Editor.

The reviewers have discussed the reviews with one another and the Reviewing Editor has drafted this decision to help you prepare a revised submission.

All reviewers share the impression that the study is comprehensive but that there are two main areas in which the manuscript required significant revisions:

1) The authors should be more explicit in explaining why they focus on FAR-A genes and;

2) In their conclusions that TEs were responsible for the proliferation of FAR. The statistical enrichment of TEs within 10kb of the target genes is weak evidence and we encourage the authors to provide a more fine grained spatial analysis (if their genomic assemblies support such an analysis) and/or temper their conclusions and provide suggestions for more rigorous tests of the TE hypothesis. A deeper discussion, as detailed below, about how the potential proliferation of the ancestral FAR-A gene by TEs should also be done.

Also, as detailed below some re-organisation of the manuscripts so that the quantification comes before the functional assay to motivate/explain the choice of substrate.

*Reviewer #1 (summary: minor comments not shown):*

This study is about FAR genes and their contribution to male-marking pheromone composition in the Hymenoptera with a focus on bumble bee species. The authors used various methods including next-generation sequencing, analysis of transposable elements (TE) in the genomic environment of FAR genes, and functional characterization of FARs in the bumble bee species *Bombus lucorum, B. lapidarius*, and *B. terrestris* to uncover what they described as a "massive expansion of the FAR gene". Functional assays were successful carried out in yeast and standard chemical analytical methods (gas chromatography and two-dimensional gas chromatography-mass spectrometry, HRMS, IR, 1D-NMR (proton and carbon) combined with organic synthesis were used to establish and confirm the identities of compounds.

In my view, this study is well executed, the transcriptomic analysis, chemistry and data analysis are generally sound, and well written and clearly presented.

*Reviewer #2 (summary: minor comments not shown):*

The reviewed manuscript by Tupec et al. with the title "Expansion of the fatty acyl reductase gene family shaped pheromone communication in Hymenoptera" exploits the diversity of fatty alcohols used by males of different bumble bee species to attract conspecific females. It does so a) to identify what genes and enzymes are involved in the biosynthesis of the above species-specific pheromone blends, b) to functionally characterize selected candidate fatty acyl reductases via heterologous expression experiments, and c) to shed light on the possible mechanisms that has caused the fatty acyl reductase gene family to be significantly more diverse in bumble bees and stingless bees than in other Hymenoptera. The study is very comprehensive and goes well beyond where most other studies end. Specifically, while there are numerous published studies that correlate gene family expansions with adaptive traits, only very few seek to functionally characterize and hence causally link gene duplications with the evolution of novel traits. Finally, the study also provides insights into how the diversity of fatty acyl reductases could have evolved from a mechanistic point of view by showing that fatty acyl reductase-encoding genes are more frequently surrounded by transposable elements that expected by pure chance. I consider the study as highly relevant and interesting for researchers in the fields of chemical communication and comparative genomics. The experiments are sound and rigorous.

My only major critic concerns the discussion on the role of TEs for the proliferation of the fatty acyl reductase-encoding genes and b) the timing of the diversification fatty acyl reductase-encoding genes. The authors speculate that "the FAR-A gene in the common ancestor of bumble bees and stingless bees translocated to a TE-rich region, which subsequently facilitated expansions of this orthology group". They mention this as one possible scenario (but do not mention alternative scenarios). Given that TEs have the ability to jump and proliferate within a genome, the possibility that TEs jumped into the vicinity of the ancestral FAR-A gene should also be discussed. Note that only a few (two!) TEs in close spatial vicinity of the ancestral FAR-A gene could explain both extant FAR-A gene diversity and TE abundance around the FAR-A gene family members. This is because the TEs could serve as seeds for ectopic recombination. Thus, the authors should also briefly discuss how the proliferation of the ancestral FAR-A gene by TEs could have taken place.

Finally, the author may want to cite the following article in this context: Schrader and Schmitz (in press). The impact of transposable elements in adaptive evolution. Mol Ecol. 2018 Jul 13. doi: 10.1111/mec.14794. Other than that, I consider the manuscript perfectly suitable for publication in *eLife*.

*Reviewer #3:*

Fatty acyl reductases (FARs) play a crucial role in the biosynthesis of fatty alcohols involved in important biological processes such as the production of wax esters, pheromone components, etc. In insects, FARs are encoded by members of a large multigene families. To date, our knowledge of the particular substrate preference and function of insect FARs remains limited to a few study systems. Tupec et al. used a multi-pronged approach to study the FARs involved in the biosynthesis of the male marking pheromones of bumblebees. Using transcriptomics and phylogenetic reconstructions, they identify a number of putative candidates for functional testing. Analyzing the genomic region surrounding FAR genes, they propose that one clade of FARs (FAR-A) might have expended due to the presence of transposable elements. Finally, combining chemical analysis and functional assays using heterologous expression in yeast, they establish a link between FAR activity and specific components found in the labial glands of male bumblebees.

All in all, this is an interesting study, well written and nicely illustrated. I only have a few specific comments, which follow.

At the moment, it may not be directly clear to the reader why FAR-A gets all the attention. In a sense, analysis of the TE-environment is one of the key experiments justifying why this particular clade may be of interest. I suggest to modify slightly this section, starting by contrasting FAR genes vs. randomly selected genes. Making individual data points for each genes would probably make appear FAR-A representative at the end of the tail. As a side note, I think that using a bar graph for this Figure (Figure 2) is a poor choice. I would prefer to see as much of the data (i.e. box-plots with individual data points visible) and give the readers a way to judge for themselves. This would make the argument stronger.

Second, I think the text would benefit from a slight reorganization. I would suggest introducing the quantification of the fatty alcohols and acyls earlier in the text, before the functional assay. That way, the choice of which substrate to test would appear more logical.

Other Comments:

No need to introduce a new hybrid chemical nomenclature: 16:1 is fine, Z9-16:OH as well, but Z9-16:1OH is not in my opinion.

Order the supplementary figures as they appear in the text.

Figure 1—figure supplement 1: indicate expression level in TPM; RPKM is a length-dependent metric, which would therefore be affected by the length of the reconstructed transcripts.

Subsection “FAR gene family evolution in Hymenoptera”: change orthologs for homologs.

Subsection “FAR gene family evolution in Hymenoptera”: Truncated sequence with missing motifs could represent an artefact of the reconstruction strategy, as there is typically not enough data to reconstruct confidently lowly expressed genes de novo.

---

## [Author Response]

All reviewers share the impression that the study is comprehensive but that there are two main areas in which the manuscript required significant revisions:1) The authors should be more explicit in explaining why they focus on FAR-A genes and;2) In their conclusions that TEs were responsible for the proliferation of FAR. The statistical enrichment of TEs within 10kb of the target genes is weak evidence and we encourage the authors to provide a more fine grained spatial analysis (if their genomic assemblies support such an analysis) and/or temper their conclusions and provide suggestions for more rigorous tests of the TE hypothesis. A deeper discussion, as detailed below, about how the potential proliferation of the ancestral FAR-A gene by TEs should also be done.Also, as detailed below some re-organisation of the manuscripts so that the quantification comes before the functional assay to motivate/explain the choice of substrate.

We have added additional FARs from other Apidae species (i.e. *A. mellifera, Euglossa dilemma, Ceratina calcarata*, and *Epeolus variegatus*) for analysis of FAR gene tree (new version of Figure 1—figfure supplement 2). We also modified the text in Results (please see the paragraph FAR gene family evolution in Hymenoptera) and in the Discussion (second paragraph). Importantly, we performed additional TE analysis in the vinicity of FAR genes of *B. impatiens*, confirming enriched TEs around FAR A genes. We prepared new version of Figure (Figure 3), added comments in Results, Discussion (sixth paragraph), and in Materials and methods (subsection “Genome alignment and TE-enrichment analysis”).

Reviewer #3:

[…] All in all, this is an interesting study, well written and nicely illustrated. I only have a few specific comments, which follow.At the moment, it may not be directly clear to the reader why FAR-A gets all the attention. In a sense, analysis of the TE-environment is one of the key experiments justifying why this particular clade may be of interest. I suggest to modify slightly this section, starting by contrasting FAR genes vs. randomly selected genes. Making individual data points for each genes would probably make appear FAR-A representative at the end of the tail. As a side note, I think that using a bar graph for this Figure (Figure 2) is a poor choice. I would prefer to see as much of the data (i.e. box-plots with individual data points visible) and give the readers a way to judge for themselves. This would make the argument stronger.

We modified the TE-analysis section in results and prepared new Figure 3, which contains individual data points for each FAR gene. We also added TE analysis results for another bumble bee species (*B. impatiens*) (Figure 3C, D).

Second, I think the text would benefit from a slight reorganization. I would suggest introducing the quantification of the fatty alcohols and acyls earlier in the text, before the functional assay. That way, the choice of which substrate to test would appear more logical.

We reordered the paragraphs.

Other Comments:No need to introduce a new hybrid chemical nomenclature: 16:1 is fine, Z9-16:OH as well, but Z9-16:1OH is not in my opinion.

Redundant nomenclature was corrected; Z9-16:1OH (and similar) were replaced with Z9-16:OH.

Order the supplementary figures as they appear in the text.

Numberingof figures was modified according to place of the first mentioning in text.

Figure 1—figure supplement 1: indicate expression level in TPM; RPKM is a length-dependent metric, which would therefore be affected by the length of the reconstructed transcripts.

Since both TPM and RPKM normalize for transcript length, we suggest to keep RPKM as adequately informative metric.

Subsection “FAR gene family evolution in Hymenoptera”: change orthologs for homologs.

Orthologs removed.

Subsection “FAR gene family evolution in Hymenoptera”: Truncated sequence with missing motifs could represent an artefact of the reconstruction strategy, as there is typically not enough data to reconstruct confidently lowly expressed genes de novo.

We agree this is most probably the case for some of the truncated FAR sequences. However, we have de novo assembled apparently truncated FAR-A sequences which are also highly abundant and for which de novo assembly is typically reliable. Additionally, sequentially highly similar sequences were recovered also from genome assemblies (see Figure 1—figure supplement 1 and Supplementary file 1). We therefore believe that this combined evidence indicates that our results or conclusions are not influenced by the inability to correctly assemble some of the low-expressed genes.